# Unpaired Multimodal Learning for Biological Datasets

**Zongliang Ji**[1,2,3]   ORCID                    JERRYJI@CS.TORONTO.EDU
**Cian Eastwood**[1]                                 CIAN@VALENCELABS.COM
**Anna Goldenberg**[2,3]                      ANNA.GOLDENBERG@UTORONTO.CA
**Paul Pu Liang**[4]                                       PPLIANG@MIT.EDU
**Jason Hartford**[1]                                 JASON@VALENCELABS.COM
**Rahul G. Krishnan**[2,3]                    RAHULGK@CS.TORONTO.EDU
**Emmanuel Noutahi**[1]                   EMMANUEL@VALENCELABS.COM

[1] *Valence Labs*
[2] *University of Toronto*
[3] *Vector Institute*
[4] *MIT*

**Editors:** Accepted for publication at MIDL 2026

**Keywords:** Unpaired Contrastive Learning, Pathology Imaging, Spatial Transcriptomics

## Abstract

Multimodal learning holds tremendous promise for biology, providing a path to integrate diverse data types and ultimately construct a more complete picture of underlying biological mechanisms. However, most existing approaches for multimodal learning require paired samples—an impractical assumption in biology, where measurement devices often destroy samples (e.g., RNA sequencing). To address this challenge, we introduce IntraPair InterCluster (IPIC), a novel contrastive approach for multimodal learning that departs from traditional reliance on paired data by requiring only treatment-group labels. IPIC aligns modalities through intra-treatment group matching and inter-treatment group clustering, producing embeddings that are both accurate and biologically meaningful. In experiments on four curated multimodal biological datasets, IPIC consistently outperforms baseline approaches, highlighting its effectiveness in leveraging independently collected single-modality datasets for multimodal contrastive pre-training.

## 1. Introduction & Related Work

Recent advances in high-throughput screening technologies have facilitated the collection of large-scale biological datasets in various modalities (Way et al., 2023; Larsson et al., 2021; Stoeckius et al., 2017; Baek and Lee, 2020; Boutros et al., 2015). As each modality alone provides a limited view, integrating them is crucial for forming a more complete picture of biology and ultimately enhancing our understanding of the underlying mechanisms (Bunne et al., 2024; Heumos et al., 2023).

In parallel, recent advances in self-supervised learning have enabled impressive multimodal capabilities in domains like computer vision and natural language processing, including zero-shot classification (Radford et al., 2021), text-to-image generation (Rombach et al., 2022; Ramesh et al., 2022), and text-based image editing (Hertz et al., 2023). However, these successes have relied on *paired* samples across modalities, such as images and their associated captions. For example, CLIP (Radford et al., 2021) uses an InfoNCE loss (Gutmann and Hyvärinen, 2010; Oord et al., 2018) to learn representations that maximize the true matching (or similarity) between representations of images and their captions.

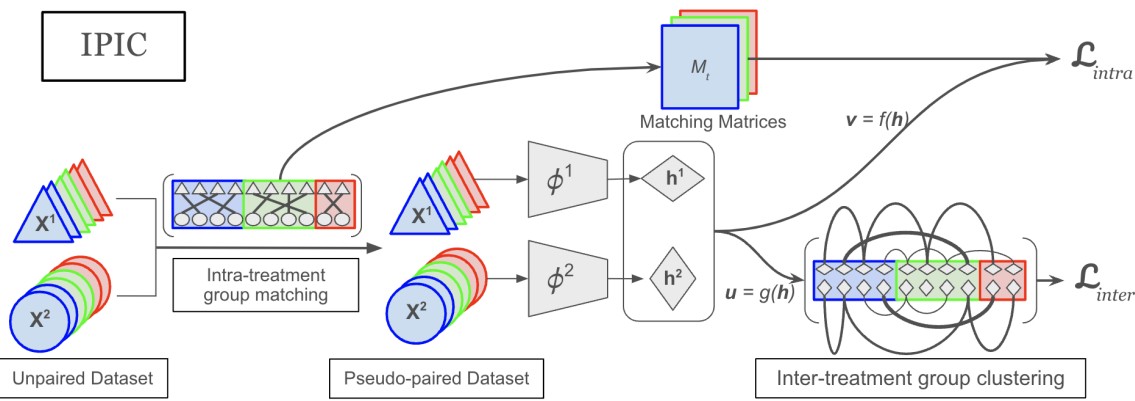

Figure 1: **Overview of our IntraPair InterCluster (IPIC) approach.** Following Algorithm 3, given an unpaired dataset (triangles and circles represent different modalities, with colors indicating treatment groups), we first perform matching within each treatment group to generate a pseudo-paired dataset. Next, we use encoders $\phi^1$ and $\phi^2$ to produce embeddings $h^1$ and $h^2$. We then apply the matching scores $M_t$ as weights with embeddings ($\mathbf{v}$) generated by the projection head $f(\cdot)$ for intra-treatment group learning using the $\mathcal{L}_{intra}$ objective (Eq. 8). Finally, we cluster embeddings ($\mathbf{u}$) generated by projection head $g(\cdot)$ to generate pseudo-labels to perform inter-treatment group learning with objective $\mathcal{L}_{inter}(Eq. 10)$.

In biology, paired samples are often difficult (if not impossible) to collect due to the *destructive nature* of many measurement devices, e.g., single-cell RNA sequencing, protein expression profiling, and high-content microscopy[1]. Thus, samples must be *indirectly linked* through shared experimental conditions, such as gene knockouts or chemical treatments. To this end, several recent studies have explored the potential of multimodal contrastive learning in the biological domain (Bao et al., 2022; Yang et al., 2022; Fradkin et al., 2024), with promising results in predicting cell states (Min et al., 2024), identifying phenotypic changes (Bao et al., 2022), and assessing perturbation outcomes (Fradkin et al., 2024). In addition, others have explored approximate matching techniques (Xi et al., 2024; Ryu et al., 2024) and anchors (Stuart et al., 2019), along with partial pairing (Zhu et al., 2023; Tu et al., 2022) and indirect links (Liu et al., 2020; Lamiable et al., 2023; Hao et al., 2021). While promising, these methods mostly assume consistent underlying correlations between modalities. However, in complex biological contexts, these correlations are often non-linear and noisy. For instance, a treatment may alter gene expression without visibly affecting cell morphology, or similar treatments may yield divergent effects across modalities. Over-reliance on such techniques can force alignments where true biological correlations may not exist, degrading the quality of learned embeddings. Our approach seeks to mitigate this by enabling more biologically relevant alignments.

The contributions of our work are as follows:

- We introduce **IntraPair InterCluster** (IPIC), a novel contrastive approach for unpaired biological datasets. IPIC aligns modalities via two complementary strategies: using treatment-group labels for intra-treatment *matching* and leveraging inherent modality structures for inter-treatment *clustering*.

---

1. See Figure 4

- Unlike previous approaches, IPIC effectively leverages both shared experimental conditions *and* intrinsic modality patterns without requiring paired samples, producing embeddings that are both accurate and biologically meaningful.
- In comprehensive experiments on four real-world "omics" datasets (phenomics and transcriptomics), we demonstrate that IPIC consistently outperforms baselines, including random pairing and weakly-supervised contrastive methods (Alwassel et al., 2020; Zheng et al., 2021), providing a foundation for extending unpaired contrastive learning to new domains.

## 2. Background

We define our unpaired problem setup before reviewing prior paired contrastive methods and their limitations in this context.

### 2.1. Problem setup: Unpaired multimodal learning

Given two treatment-labeled datasets $\mathcal{D}_1 = \{(x_i^1, t_i)\}_{i=1}^{N_1}$ and $\mathcal{D}_2 = \{(x_j^2, t_j)\}_{j=1}^{N_2}$, where $x_i^1 \in X^1$ (e.g., images) and $x_j^2 \in X^2$ (e.g., sequences or texts) are samples from different modalities, each sample is associated with a treatment label $t \in T$ drawn from the set of possible treatments $T$ (see Figure 2). $\mathcal{D}_1$ and $\mathcal{D}_2$ share the same set of treatment labels, allowing us to construct a combined dataset $\mathcal{D} = \{(x_i^1, x_j^2, t)\}_{i,j=1}^{N}$. However, samples $x_i^1$ and $x_j^2$ are not directly paired (i.e., they do not correspond to the same cell or natural scene), but are instead indirectly linked by their shared treatment label $t$. As in Figure 1, our goal is to learn encoders $\phi^1, \phi^2$ that produce improved representations, measured by zero-shot treatment prediction and downstream task performance.

### 2.2. Contrastive learning for *paired* data

Standard methods like SimCLR (Chen et al., 2020) and CLIP (Radford et al., 2021) align paired samples $\{(x_i^1, x_i^2)\}_{i=1}^{B}$ by embedding them into vectors $\mathbf{v}$ via encoders $\phi$ and heads $f$. The InfoNCE (Gutmann and Hyvärinen, 2010; Oord et al., 2018) loss maximizes similarity between true pairs: $\mathcal{L}_{NCE}^{i(1)} = -\log \frac{\exp(\text{sim}(\mathbf{v}_i^1, \mathbf{v}_i^2)/\tau)}{\sum_{l=1}^{B} \mathbb{1}_{[l \neq i]} \exp(\text{sim}(\mathbf{v}_i^1, \mathbf{v}_l^2)/\tau)}$ (1). The total objective averages over both modalities $\mathcal{L}_{NCE} = \frac{1}{2B} \sum_i^B (\mathcal{L}_{NCE}^{i(1)} + \mathcal{L}_{NCE}^{i(2)})$ (2)[2]. In our unpaired setting, where batches $\{(x_i^1, x_j^2)\}_{i,j=1}^{B}$ share only treatment labels ($t_i = t_j$), standard InfoNCE forces arbitrary alignment of same-treatment samples.

Weak supervision from treatment labels allows for variants like Supervised Contrastive Learning (SupCon (Khosla et al., 2020)) which aligns all pairs within a treatment group: $\mathcal{L}_{SupCon}^{i(1)} = -\sum_j^B \log \frac{\mathbb{1}_{[t_i = t_j]} \cdot \exp(\text{sim}(\mathbf{v}_i^1, \mathbf{v}_j^2)/\tau)}{\sum_{l=1}^{B} \mathbb{1}_{[l \neq j]} \exp(\text{sim}(\mathbf{v}_i^1, \mathbf{v}_l^2)/\tau)}$ (3). InfoCore (Wang et al., 2024) combines these: $\mathcal{L}_{InfoCore} = \frac{1}{2} \mathcal{L}_{SupCon} + \frac{1}{2} \mathcal{L}_{NCE}$ (4).

Alternatively, XDC (Alwassel et al., 2020) and WCL (Zheng et al., 2021) use cluster-derived weak labels $c$ from representations to guide learning: $\mathcal{L}_{WCL}^{i(1)} = -\sum_j^B \log \frac{\mathbb{1}_{[c_i^2 = c_j^2]} \cdot \exp(\text{sim}(\mathbf{v}_i^1, \mathbf{v}_j^2)/\tau)}{\sum_{l=1}^{B} \mathbb{1}_{[l \neq j]} \exp(\text{sim}(\mathbf{v}_i^1, \mathbf{v}_l^2)/\tau)}$ (5). However, none of these methods effectively handle the unpaired biological setting, as even SupCon lacks mechanisms to distinguish specific instances within the same treatment group.

### 2.3. Learning from *unpaired* data

Unpaired learning often employs cross-modal translation (Nakada et al., 2023; Park et al., 2020; Sturma et al., 2024), cycle consistency (Zhu et al., 2017; Almahairi et al., 2018; Amodio and Krishnaswamy, 2018; Tsai et al., 2022), or Optimal Transport (Demetci et al., 2022; Ryu et al., 2024; Gao

---

2. We refer only to the $i$th example of modality 1 for simplicity; the total is averaged over both directions.

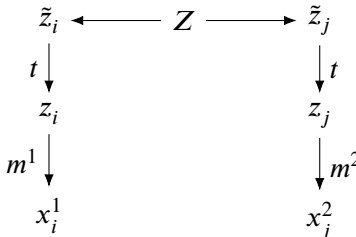

Figure 2: **Data generating process for unpaired biological datasets.** $Z$ represents the original sample distributions (e.g. HUVEC cells (Baudin et al., 2007)). Two samples $\tilde{z}_i, \tilde{z}_j$ are drawn from this distribution and have the same treatment $t$ applied, resulting in treated samples $z_i = t(\tilde{z}_i)$ and $z_j = t(\tilde{z}_j)$. Finally, measurement devices $m^1, m^2$ are used to get per-modality observations $x_i^1 = m^1(z_i)$ and $x_j^2 = m^2(z_j)$ sharing the same treatment $t$.

---

**Algorithm 1:** Matching and Re-pairing

---

**Input:** Unpaired dataset $\mathcal{D} = \{(x_i^1, x_j^2, t)\}_{i,j=1}^N$

**Output:** Matching matrices $\{M_t\}$, pseudo-paired $\mathcal{D}' = \{(x_i^1, x_k^2, t)\}$

1 Train $\psi^1, \psi^2$ on $\{(x_i^1, t_i)\}, \{(x_j^2, t_j)\}$; compute propensity scores $\{\pi_i^1\}, \{\pi_j^2\}$

2 **for** *each treatment t in T* **do**

3      Filter samples by $t$; compute cost $C_{ij} = \|\pi_i^1 - \pi_j^2\|_2$ and set uniform $p_1, p_2$

4      Solve EOT via Sinkhorn to get $M_t$: $\min_M \sum C_{ij} M_{ij} - \lambda H(M)$ s.t. $M\mathbb{1}=p_1, M^T\mathbb{1}=p_2$

5      Reorder modality 2 samples using $M_t$ to form pairs $\{(x_i^1, x_k^2)\}$

---

et al., 2020; Kriebel and Welch, 2022). However, biological data's inherent noise and non-linearity limit cycle consistency, and OT relies on metric assumptions that often fail in this domain. We focus specifically on adapting representation learning to these biological constraints.

## 3. Methods

We now introduce our proposed method, *IntraPair InterCluster* (IPIC), which leverages both intra-treatment group information via matching (or pairing) *and* inter-treatment group information via clustering.

### 3.1. Intra-treatment Group Learning via Matching

To address the challenge of unpaired contrastive learning more effectively, we first explore using matching methods by leveraging intra-treatment group information to align samples across modalities. This involves re-pairing samples across the two modalities by computing matching scores between each sample in one modality and all samples in the other modality that share the same treatment label. We refer to this step as *intra-treatment group learning*.

To leverage the treatment group label and allow our encoders to distinguish samples within each group, we start by re-pairing the two modalities within the same treatment group. This draws inspiration from causal inference methods which match unpaired modalities with shared latent features (Xi et al., 2024; Ryu et al., 2024). As illustrated in Figure 2, both modalities are assumed to capture measurements from a shared latent space $Z$ (e.g. natural scenes or identical cell lines). Thus each modality, $X^1$ or $X^2$, can be viewed as a *partial* observation of $Z$. Ideally, if we could observe the corresponding latent variables $z_i$ and $z_j$, re-pairing could be straightforward since we could directly match the samples $x_i^1$ and $x_j^2$ based on their proximity in this shared space. However, since $Z$ is un-

---

**Algorithm 2:** Cluster Pseudo-labeling

---

**Input:** Batch $\mathcal{B} = \{(\mathbf{u}_i^1, \mathbf{u}_k^2)\}$, number of treatments $|T|$
**Output:** Pseudo-labels $\{c_i^1\}$ and $\{c_k^2\}$
1 Run *KMeans* ($K = |T|$) on $\{\mathbf{u}_i^1\}$ and $\{\mathbf{u}_k^2\}$ separately to obtain labels $\{c_i^1\}$ and $\{c_k^2\}$

---

observable and difficult to estimate without introducing unverifiable assumptions, we must rely on alternative approaches to match $x_i^1$'s and $x_j^2$'s effectively. To approximate this latent alignment, we use propensity scores as surrogates for the latent features, enabling us to compute similarity scores between samples across modalities.

Following Xi et al. (2024), we frame the matching problem using principles from causal inference (Rubin, 1974). Specifically, we assume that each treatment $t$ induces perturbations through a shared latent space $Z$, enabling us to use estimated propensity scores, $p(t|Z)$, as a proxy for the unobserved latent space $Z$. Additionally, we assume that the conditional distributions $t|X^1$ and $t|X^2$ are identical, such that $t|X^1 \stackrel{d}{=} t|X^2 \stackrel{d}{=} t|Z$. Under these assumptions, propensity scores can serve as a common space for aligning modalities, making them fully identifiable through classifiers trained independently on each modality. This enables us to estimate propensity scores solely from observational data [3].

With the above setup, given $\mathcal{D} = \{(x_i^1, x_j^2, t)\}_{i,j=1}^N$, our goal is to compute a matching matrix $M_t$ for each $t \in T$ to re-pair samples across the two modalities within each treatment group based on $M_t$. We begin by training classifiers (propensity score predictor) $\psi^1$ and $\psi^2$ for each modality using $\{x_i^1, t_i\}$ and $\{x_j^2, t_j\}$. This yields propensity scores $\{\pi_i^1\}$ and $\{\pi_j^2\}$, where $\pi \in \mathbb{R}^{|T|}$. For each unique treatment group $t$, we then apply entropic optimal transport to obtain a matching matrix $M_t$, which enables a new pairing within each treatment group. The result is a pseudo-paired dataset $\mathcal{D}' = \{(x_i^1, x_k^2, t)\}_{i,j=1}^N$, where modality 2 is now indexed by $k$. The detailed process for re-pairing and obtaining matching matrices is outlined in Algorithm 1.

After obtaining the pseudo-paired dataset and matching matrix for each treatment group $t$, we define our *intra-treatment group* learning objective as follows:

$$\mathcal{L}_{intra}^{i(1)} = -\sum_k^B \log \frac{\mathbb{1}_{[t_i=t_k]} \cdot m_{i,k}^1 \cdot \exp(\text{sim}(\mathbf{v}_i^1, \mathbf{v}_k^2)/\tau)}{\sum_{l=1}^B \mathbb{1}_{[l \neq k]} \exp(\text{sim}(\mathbf{v}_i^1, \mathbf{v}_l^2)/\tau)}, \tag{6}$$

where $\sum_{k=1}^B m_{i,k}^1 = 1$ represents the adjusted matching scores between each sample $i$ and all of its cross-modal pairs within the sampled batch. The matching score can be defined as

$$m_{i,k}^1 = \begin{cases} (M_{t_i})_{(a,b)}, & \text{if } t_i=t_k, \text{ and } (a,b) \text{ are indices of } i,k \text{ in } t \\ 0, & \text{otherwise} \end{cases} \tag{7}$$

This ensures that $m_{i,k}^1$ is non-zero only when both samples $i$ and $k$ belong to the same treatment group $t$ and are aligned according to the matching matrix $M_t$. The full *intra-treatment group* objective becomes: $\mathcal{L}_{intra} = \frac{1}{2B} \sum_{i,k}^B (\mathcal{L}_{intra}^{i(1)} + \mathcal{L}_{intra}^{k(2)})$ (8), and enables the encoders to learn treatment-aware information while distinguishing sample pairs across modalities.

### 3.2. Inter-treatment Group Learning via Clustering

In addition to *intra-treatment group* learning, a robust approach for unpaired contrastive learning should account for cases where different treatment groups yield similar or no effect at all on certain

---

3. The complete theoretical setup and assumptions of the matching problem are provided in the supplement.

---

**Algorithm 3:** Batch IntraPair InterCluster Contrastive Learning (IPIC)

---

**Input:** Unpaired dataset $\mathcal{D}$, loss scale $\lambda$, encoders $\phi^{1,2}$, heads $f^{1,2}, g^{1,2}$

1   Estimate matching matrices $\{M_t\}$ and pseudo-paired $\mathcal{D}'$ via Alg. 1

2   **for** *epoch* = 1 **to** *E and step* = 1 **to** *S* **do**

3      Sample batch $\mathcal{B}$ from $\mathcal{D}'$ ($|\mathcal{B}| = B$)

4      Compute and normalize matching scores $m^1, m^2$ for all pairs in $\mathcal{B}$ using Eq. (7)

     /* Forward pass for both views $v \in \{1, 2\}$                                  */

5      Compute $\mathbf{h}^v = \phi^v(x^v)$, $\mathbf{v}^v = f^v(\mathbf{h}^v)$, and $\mathbf{u}^v = g^v(\mathbf{h}^v)$

6      Calculate $\mathcal{L}_{intra}$ using $\{m, \mathbf{v}\}$ (Eq. 8)

7      Get cluster pseudo-labels $\{c^1, c^2\}$ via Alg. 2

8      Calculate $\mathcal{L}_{inter}$ using $\{c, \mathbf{u}\}$ (Eq. 10)

9      Optimize $\phi^{1,2}, f^{1,2}, g^{1,2}$ to minimize $\mathcal{L}_{intra} + \lambda \mathcal{L}_{inter}$

---

samples. In biological datasets, it is common for some treatments to have minimal or overlapping effects, which can lead to similar representations across different treatment groups (Fradkin et al., 2024). If two samples in modality 2, $x_j^2$ and $x_k^2$, exhibit similar traits in the original population (i.e., their latent variables $z_j$ and $z_k$ are similar) and receive treatments $t_1$ and $t_2$ with no substantial effect, then their representations $\mathbf{v}_j^2$ and $\mathbf{v}_k^2$ should remain similar. In such cases, $\mathbf{v}_j^2$ and $\mathbf{v}_k^2$ should both be considered as positive pairs with their cross-modality counterparts (i.e. $\mathbf{v}_i^1$). Motivated by this observation, we incorporate clustering in the representation space to enable *inter-treatment group* learning.

To capture these inter-treatment similarities, we introduce a clustering mechanism that assigns pseudo-labels to the embeddings of each modality. Inspired by methods like XDC (Alwassel et al., 2020) and WCL (Zheng et al., 2021), we add a projection head $g$ to our network specifically designed for clustering to produce embeddings $\{\mathbf{u}^\blacksquare\} = g^\blacksquare(\phi^\blacksquare(\{x^\blacksquare\}))$. The goal is to leverage cluster assignments from one modality to guide the learning process for the other modality, thereby enhancing cross-modal alignment that generalize across treatments. This clustering process is outlined in Algorithm 2.

With these generated pseudo-labels, we define our *inter-treatment group* objective as:

$$\mathcal{L}_{inter}^{i(1)} = -\sum_o^B \log \frac{\mathbb{1}_{[i=k]} \cdot \mathbb{1}_{[c_k^2 = c_o^2]} \cdot \exp(\text{sim}(\mathbf{u}_i^1, \mathbf{u}_o^2)/\tau)}{\sum_{l=1}^B \mathbb{1}_{[l \neq k]} \exp(\text{sim}(\mathbf{u}_i^1, \mathbf{u}_l^2)/\tau)}, \tag{9}$$

The complete inter-treatment group objective is defined as: $\mathcal{L}_{inter} = \frac{1}{2B} \sum_{i,k}^B (\mathcal{L}_{inter}^{i(1)} + \mathcal{L}_{inter}^{k(2)})$ (10), It enables the model to identify and bring together similar samples across different treatment groups, thereby capturing nuanced biological patterns and enhancing the robustness of our unpaired contrastive learning framework.

### 3.3. Combined algorithm

We now combined our *intra-treatment group* and *inter-treatment group* objectives to form our unified IPIC method for unpaired multimodal representation learning on biological datasets. Figure 3 illustrates a conceptual comparison between IPIC and baseline methods, highlighting their differences. Our method begins by re-pairing the dataset using Algorithm 1 to generate matching matrices for each treatment group. For each batch sampled from the dataset, we retrieve the corresponding matching scores for each sample in each modality via the matching matrix. This process is implemented with a custom collate function in the data loader, ensuring efficient score retrieval and normalization

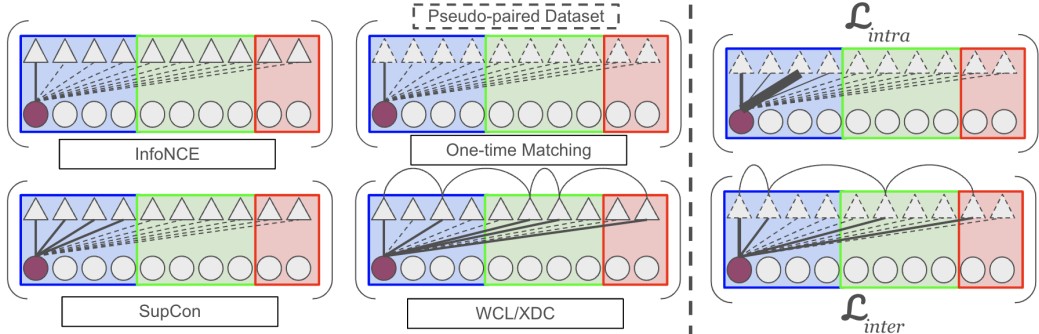

Figure 3: **Method comparison.** This figure shows the various methods we experimented with, where different shapes represent different modalities, and different colors represent different treatment groups. Each figure demonstrates how the method objective is computed within a batch for a specific sample (highlighted circle) in one modality. Solid lines connecting circles and triangles indicate positive pairs, while dashed lines indicate negative pairs. Dashed triangles represent the pseudo-paired dataset generated by matching within each treatment group (Algorithm 1). In $\mathcal{L}_{intra}$, the solid lines have varying widths to denote different weights applied to each positive pairing based on the matching score. In WCL (Zheng et al., 2021), XDC (Alwassel et al., 2020), and $\mathcal{L}_{intra}$, curved lines connect triangles that share the same cluster label. With our $\mathcal{L}_{inter}$ (Eq. 10) and $\mathcal{L}_{intra}$ (Eq. 8) design, each sample can learn both intra- and inter-treatment group information.

to prevent numerical errors. We also employ two distinct sets of projection heads: one for *intra-treatment group* representation ($f^1, f^2$) and one for *inter-treatment group* representation ($g^1, g^2$). This separation prevents conflicts within the embedding space between intra- and inter-treatment group learning objectives. Finally, we optimize the network using the complete objective. The full steps of IPIC are outlined in detail in Algorithm 3.

## 4. Experiments

### 4.1. Datasets

Our benchmarks consist of two modalities ($X^1, X^2$), distinct treatments ($T$), and orthogonal downstream labels ($Y$) related to the latent distribution; detailed descriptions and generation protocols are provided in Appendix A.

**Bio-Augmented IRFL (BIRFL):** Derived from IRFL (Yosef et al., 2023), we applied seven augmentations ($T$) to 786 image-text pairs, introducing random noise to mimic biological variability. This results in 5,502 unique pairs of images ($X^1$) and captions ($X^2$), with $Y$ representing figurative types. **Spatial Transcriptomics (ST):** Sourced from Zhou et al. (2022), this dataset contains 57,363 paired pathology image patches ($X^1$) and RNA readouts ($X^2$) from 15 pancreatic cancer biopsies. $T$ denotes one of four chemotherapy regimens, and $Y$ indicates the presence of tumor tissue. **Compound Treated (COMP):** An unpaired dataset of HUVEC cells (Baudin et al., 2007) treated with 78 bioactive compounds ($T$). We randomly paired 36,562 samples consisting of cell-painting crops ($X^1$) (Bray et al., 2016; Fay et al., 2023) and bulk RNA-seq data ($X^2$) (Ye et al., 2018) sharing the same compound. $Y$ denotes solution concentration. **Gene Knockout (GKO):** Similar to COMP,

Table 1: Test Set Results on Four Curated Datasets. We report accuracy for Treatment and Task label predictions. **Bold** indicates best performance.

| | BIRFL | | ST | | COMP | | GKO | |
|---|---|---|---|---|---|---|---|---|
| Method | Treat. | Task | Treat. | Task | Treat. | Task | Treat. | Task |
| *Reference Baselines* | | | | | | | | |
| Raw Features ($x^1 || x^2$) | 77.0 | 55.4 | 77.7 | 63.6 | 74.3 | 58.6 | 19.9 | 28.9 |
| InfoNCE-paired (Oracle) | 81.4 | 97.2 | 98.5 | 80.9 | - | - | - | - |
| FactorCL-SSL(Liang et al., 2024) | 80.7 | 64.5 | 92.1 | 67.3 | 90.9 | 72.5 | 25.1 | 37.9 |
| FactorCL-Sup(Liang et al., 2024) | 81.0 | 66.5 | 95.7 | 68.9 | 91.6 | 73.4 | 30.7 | 35.2 |
| InfoNCE-unpaired | 81.4 | 64.7 | 97.0 | 68.4 | 95.6 | 72.9 | 22.6 | 36.9 |
| SupCon(Khosla et al., 2020) | 83.0 | 65.3 | **99.4** | 68.9 | 96.7 | 73.4 | **34.1** | 33.7 |
| InfoCore(Wang et al., 2024) | 82.1 | 65.2 | 98.8 | 68.4 | 96.3 | 73.6 | 27.8 | 34.9 |
| WCL(Zheng et al., 2021) | 81.1 | 67.4 | 96.6 | 68.8 | 96.4 | 75.6 | 23.2 | 36.8 |
| XDC(Alwassel et al., 2020) | 80.8 | 67.2 | 94.7 | 68.4 | 96.2 | 77.6 | 21.8 | 37.3 |
| *Matching Strategy Ablation* | | | | | | | | |
| 1X-Matching (SNN) | 74.2 | 65.2 | 93.9 | 68.2 | 91.1 | 74.2 | 19.7 | 37.5 |
| 1X-Matching (OT) | 81.5 | 66.7 | 95.9 | 68.6 | 95.7 | 73.0 | 23.6 | 36.6 |
| Ours (intra-SNN) | 73.9 | 69.4 | 96.5 | 68.0 | 92.4 | 76.1 | 20.7 | 41.3 |
| Ours (intra-OT) | 83.4 | 69.7 | 97.8 | 70.4 | 97.4 | 78.5 | 26.0 | 42.8 |
| *Clustering Ablation (IPIC)* | | | | | | | | |
| Ours (intra+inter, $K = 0.25|T|$) | - | - | - | - | 97.3 | 80.1 | 25.6 | 39.4 |
| Ours (intra+inter, $K = 0.5|T|$) | 82.3 | 66.4 | 96.7 | 70.0 | 97.4 | **80.3** | 25.4 | 43.0 |
| Ours (intra+inter, $K = 2|T|$) | **83.5** | 69.3 | 97.0 | 70.1 | 97.3 | 79.7 | 23.6 | 42.0 |
| Ours (intra+inter, $K = 4|T|$) | 82.8 | 69.2 | 96.9 | 70.0 | 97.2 | 79.0 | 23.7 | 42.2 |
| **Ours (intra+inter, $K = |T|$)** | **83.5** | **71.5** | 97.7 | **70.7** | **97.8** | 80.2 | 26.4 | **43.3** |

this dataset links HUVEC cells subjected to 25 CRISPR knockouts (Bock et al., 2022) ($T$). We associated 59,011 cell-painting crops ($X^1$) (Bray et al., 2016) with perturb-seq profiles ($X^2$) (Dixit et al., 2016) via shared treatments. $Y$ represents binned total gene counts.

## 4.2. Experimental Setup

To simulate the unpaired setting for the naturally paired datasets (BIRFL and ST), we shuffled samples within treatment groups in the training set to create $\mathcal{D}$. For the unpaired datasets (COMP and GKO), we used manual matching in the test set solely for evaluation purposes. To ensure fair comparison, all samples were embedded using large pre-trained models (Kraus et al., 2024; Chen et al., 2024; Dosovitskiy, 2020; Lopez et al., 2018) before training (dimensions detailed in Table 4).

We employed a consistent CLIP-like architecture (Radford et al., 2021) across all methods, consisting of modality-specific vector encoders and projection heads tailored to each objective. Models were trained using their respective contrastive losses, and the resulting encoders were frozen to generate test-set embeddings. We evaluate performance on both treatment classification and downstream tasks using logistic regression on the concatenated embeddings. All experiments were repeated with eight random seeds for robustness. Further details are provided in Appendix B.

## 5. Results

We present a comprehensive evaluation of our proposed IPIC method against baselines tailored for paired datasets, weakly-supervised contrastive frameworks, and matching-based alignment. The

zero-shot performance of test set embeddings, evaluated on treatment and downstream task prediction, is summarized in Table 1 [4].

**Paired Assumptions vs. Unpaired Reality:** We first examined the impact of exact pairing versus random pairing. The BIRFL and ST datasets, which possess true pairings, allow us to test performance degradation when precise pairings are removed. As shown in the first two rows of Table 1, downstream task performance drops significantly when using randomly shuffled pairs within treatment groups. We also evaluated **FactorCL** (SSL and Sup (Liang et al., 2024)) as an additional baseline; its performance aligned closely with InfoNCE-unpaired, further confirming that methods assuming paired modalities struggle to recover representation quality when restricted to group-level alignment. This highlights that treating randomly paired samples as genuinely "paired" yields suboptimal embeddings.

**Robust Performance over Baselines:** We compared IPIC with six baseline methods, including standard InfoNCE (Eq.2), treatment-guided methods like SupCon (Eq.3) and InfoCore, and clustering-based methods WCL (Eq. 5) and XDC (Alwassel et al., 2020). Additionally, we provide linear probing results on **raw input vectors** $(x^1||x^2)$ to establish a performance lower bound. As shown in Table 1, IPIC consistently outperforms all learned baselines and significantly surpasses the raw feature baseline (e.g., on BIRFL Task prediction), demonstrating that our method learns non-trivial, biologically meaningful structure rather than simply retaining input statistics. Notably, IPIC rivals fully supervised approaches like SupCon in treatment prediction while achieving superior downstream task performance.

**Dynamic Alignment & Matching Strategy:** A key contribution of IPIC is its dynamic pairing mechanism. We compared our iterative approach against a static baseline, "1X-Matching", where Algorithm 1 is applied only once before training. IPIC achieves a **4.8% average improvement** on downstream tasks compared to one-time matching (Xi et al., 2024), confirming that iterative propensity score updates better capture underlying biological mechanisms where treatments induce similar population-level effects despite individual-cell heterogeneity. Furthermore, we ablated the matching algorithm itself, comparing our Optimal Transport (OT) approach inspired by (Xi et al., 2024) against Shared Nearest Neighbor (SNN) matching. We found OT to be consistently superior (e.g., SNN degrades performance by ~7-10% on BIRFL), likely because OT enforces global distributional constraints that are more robust to batch effects than local neighbor-based methods.

**Clustering Granularity ($K$):** For the inter-treatment clustering objective, we investigated the sensitivity of the number of clusters $K$. We evaluated $K \in \{0.25|T|, 0.5|T|, |T|, 2|T|, 4|T|\}$ and found that alternatives did not outperform $K = |T|$. This is intuitive, as setting $K$ equal to the number of treatment groups ensures that clusters align with the experimental design, preserving biological relevance while allowing the model to bridge similar treatment effects.

**Single-Modality vs. Multi-Modality:** To assess the benefits of data integration, we compared concatenated multi-modality embeddings against single-modality embeddings obtained from learned encoders $\phi^1$ and $\phi^2$ on downstream task $Y$. As shown in Table 8, multi-modality training proves consistently beneficial; with the exception of the ST image modality, concatenated embeddings outperform single-modality counterparts across all datasets. Specifically, relying on a single modality results in accuracy drops ranging from **4% to 26%**. Furthermore, a comparison with baselines in Table 1 highlights that IPIC consistently improves the quality of *each* single-modality embedding. This

---

4. See full results with standard deviation in Table 7.

demonstrates that our unpaired contrastive framework effectively leverages cross-modal synergy to enhance representation learning, benefiting both joint and individual modality inference.

**Impact of Treatment Group Labels:** While InfoNCE achieves similar treatment prediction accuracy for paired and unpaired datasets (as unpaired samples are simply shuffled within groups), incorporating treatment labels significantly improves representation quality. Methods explicitly leveraging treatment labels (SupCon, InfoCore, Matching, IPIC) consistently outperform those that do not (InfoNCE, WCL, XDC) on treatment prediction. This supports our hypothesis that shared treatment information acts as a strong training signal, which is particularly valuable in biological contexts like drug response prediction where capturing subtle treatment effects is crucial (Fradkin et al., 2024; Iorio et al., 2016).

**Orthogonality of Treatment and Downstream Tasks:** We consistently observed that baselines excelling in treatment label prediction often underperformed on downstream tasks. This suggests that our downstream task labels capture latent information intrinsic to the original sample distribution $Z$, independent of treatment labels $T$. Consequently, strong performance on downstream tasks serves as a robust indicator of representation quality, reflecting the model's ability to capture biologically meaningful variations beyond simple treatment effects. IPIC excels here by leveraging treatment labels for alignment while preserving the rich latent features required for downstream tasks.

**Ablation Study on IPIC Objectives:** Finally, we compared two IPIC variants: one using only the intra-treatment objective ($\mathcal{L}_{intra}$) and another combining both intra- and inter-treatment objectives ($\mathcal{L}_{intra} + \mathcal{L}_{inter}$). Leveraging just the intra-treatment objective yields significant gains over baselines by enabling weighted positive pairings. Adding inter-treatment clustering further boosts performance by aligning samples not only with cross-modal pairs but also with similar representations across different treatment groups. This dual strategy enables the learning of richer, more robust representations.

## 6. Conclusion

In this paper, we presented IntraPair InterCluster (IPIC), a contrastive approach for multimodal learning on biological datasets without paired samples. By combining intra-treatment matching (or pairing) and inter-treatment clustering, IPIC produces biologically-meaningful embeddings using only treatment-group labels. Our experiments on four unpaired multimodal datasets demonstrated that IPIC consistently outperformed baseline methods designed for paired data, highlighting its potential for biological applications where destructive measurement techniques often prevent the collection of paired samples. Looking ahead, IPIC's flexibility opens opportunities in domains where direct pairings between modalities are impractical. Addressing challenges like noisy, high-dimensional data and incorporating domain-specific priors could further expand its impact, driving insights across fields from healthcare to environmental science and personalized medicine.

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

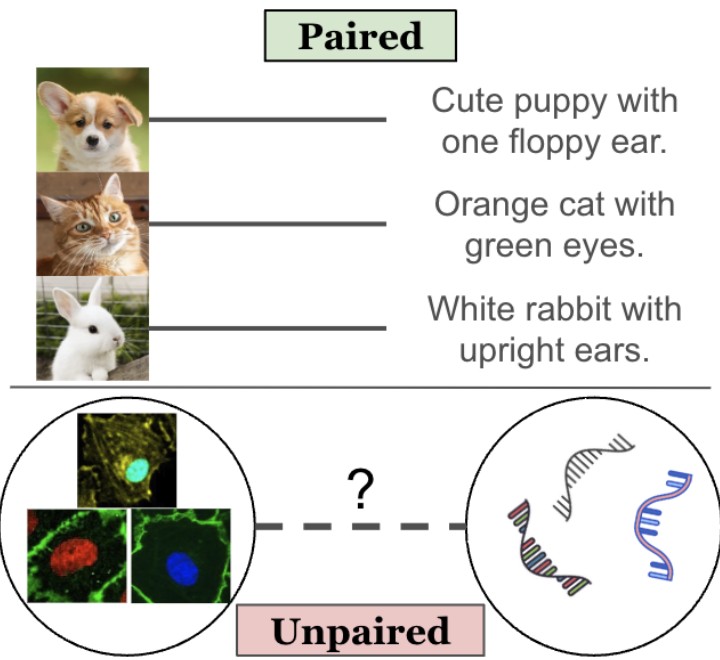

Figure 4: **Paired image-text data vs. unpaired biological data.** Unlike natural image-text pairs, biological data collection is destructive, resulting in unpaired modalities. Here, phenomics (cell imaging) and transcriptomics (gene sequencing) data cannot be collected from the same sample, leading to unpaired datasets.

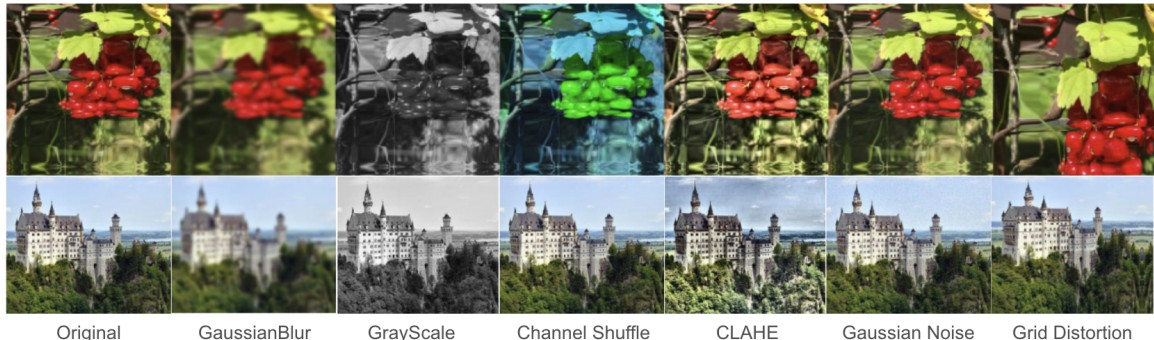

Figure 5: BIRFL Treatments and Sample illustration.

## Appendix A.  Detailed Dataset Description

In this section, we provide a comprehensive description and visualizations of the four curated datasets used in our study.

### A.1.  Bio-Augmented IRFL Dataset (BIRFL)

To assess whether shuffling within treatment groups impacts embedding or representation performance on downstream tasks, we first tested this hypothesis using a dataset with true paired information. Specifically, we extended the Image Recognition of Figurative Language (IRFL) dataset (Yosef et al., 2023), which consists of 786 paired social media images and their corresponding captions. Each pair is labeled with a figurative type: *simile*, *metaphor*, or *idiom*.

To create treatment groups $T$, we applied seven augmentation methods to the images: ***Identical Mapping***, ***Gaussian blur***, ***Gray Scale Conversion***, ***Channel Shuffle***, ***CLAHE (Contrast Limited Adaptive Histogram Equalization)***, ***Gaussian Noise Addition***, and ***Grid Distortion***. Each augmentation was paired with a descriptive modification to the corresponding caption. For example, if an image was augmented with ***Gaussian Noise Addition***, the associated caption would add the sentence, *"this image has gaussian noise added."*

To further increase the variability of the text augmentation, we generated 10 additional rephrased versions of each descriptive sentence for every augmentation method. For instance, rephrased descriptions for Gaussian noise included:

0. *"this image has gaussian noise added."*
1. *"gaussian noise has been added to this image."*
2. *"this picture includes gaussian noise."*
3. *"the image has been modified with gaussian noise."*
4. *"gaussian noise has been introduced to this image."*
5. *"this image now features added gaussian noise."*
6. *"the picture has undergone the addition of gaussian noise."*
7. *"gaussian noise has been incorporated into this image."*
8. *"the image contains added gaussian noise."*
9. *"gaussian noise has been applied to this image."*
10. *"this image has been augmented with gaussian noise."*

For each augmented image, one rephrased sentence was randomly selected and appended to the corresponding caption. This process ensured that the paired data remained consistent across modalities while introducing diverse treatment effects in both visual and textual representations. A visualization of these augmentations is provided in Figure 5.

To emulate the complexity of biological data generation, we introduced additional variability to the augmentations for each image-text pair. Within each treatment group, originally comprising 786 image-text pairs, we applied the following modifications to subsets of the data to simulate diverse treatment outcome effects:

- **No Treatment Effect:** For 10% of the samples, we applied no augmentation to mimic a control or no-treatment scenario.
- **Same Treatment Effect for Different Treatments:** Another 10% of the samples were randomly assigned a treatment augmentation other than the intended one, simulating overlap in treatment effects.
- **Mixed/Combined Treatment Effect:** For 10% of the samples, we applied a combination of the correct augmentation and an additional randomly selected augmentation to mimic the effect of mixed or combined treatments, reflecting the complexity observed in biological datasets.

This bio-inspired augmentation process resulted in the creation of seven treatment groups derived from the original 786 image-text pairs, culminating in our Bio-Augmented IRFL dataset, which contains a total of 5,502 unique image ($X^1$) and text caption ($X^2$) pairs.

To facilitate experiments for comparing various contrastive learning objectives, we embedded each image and text caption into vector spaces using state-of-the-art models. Images were embedded using the Vision Transformer (ViT-L-16) model (Dosovitskiy, 2020) (pretrained encoder $\mathcal{F}_1$), resulting in vectors of dimension 786. Text captions were embedded using the SFR-2 model (Lee et al., 2024) ($\mathcal{F}_2$), resulting in vectors of dimension 4,096.

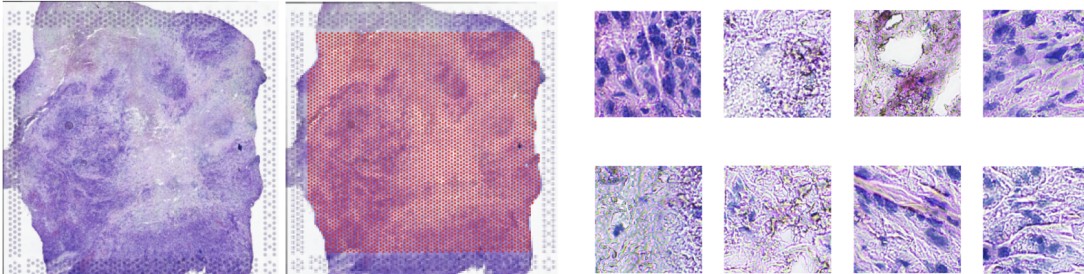

Figure 6: **Patient Biopsy Spatial Transcriptomics Data.** Our ST dataset consists of spatial transcriptomics data from patients diagnosed with pancreatic cancer. Each small tissue sample is scanned using microscopy and sequencing machines, resulting in histopathology images with various sites marked as red dots. Each site represents a sample in our ST dataset, where we obtain both the RNA-sequence readout and the corresponding image patch (right).

The paired structure of the BIRFL dataset allows us to rigorously evaluate contrastive learning methods for paired data as well as their performance with unpaired data using only the weak labels associated with treatments.

## A.2. Spatial Transcriptomics for Pancreatic Cancer Tumor Biopsy (ST)

Our second curated dataset is a biological dataset of spatial transcriptomics (ST) data derived from biopsies of patients diagnosed with Pancreatic Ductal Adenocarcinoma (PDAC), as released by Zhou et al. (2022). PDAC is a lethal disease with limited treatment options and poor survival rates. The released dataset includes samples from 31 patients, of which only 15 spatial transcriptomics (ST) samples are publicly available. Each ST sample is scanned using 10x Visium technology and is represented by a histopathology slide along with a gene-expression matrix. In this matrix, each row corresponds to RNA expression data indexed by the spatial location of the histopathology slide. A visualization of the data samples is shown in Figure 6.

Thus, ST data forms a naturally paired biological dataset, where each spatial location on the histopathology slide has a corresponding pathology image patch ($X^1$) and an RNA-seq readout ($X^2$) from the same tissue region. Since biopsies do not consist solely of tumor cells, standard practice involves experienced pathologists annotating histopathology slides, identifying different regions and cell types. For simplicity, we define our downstream task as a binary classification problem: predicting whether a site corresponds to a pathologist-labeled tumor region ($Y$). Since each slide represents a biopsy from a single patient, we define treatments ($T$) as one of four options received by the patient before the biopsy was taken: *FOLFIRINOX*, *Naive*, *Chemo-RT*, or *Mixed*.

In this dataset, each spatial site with an image-expression pair is treated as a sample, resulting in 57,363 image-expression pairs from the 15 ST samples. For each image patch, we use the pathology foundation model UNI (Chen et al., 2024) (pretrained encoder $\mathcal{F}_1$) to embed the image into a 1024-dimensional vector. Similarly, we train an scVI (Single-cell Variational Inference) model (Lopez et al., 2018) ($\mathcal{F}_2$) on all 57,363 RNA expression profiles, embedding each into a 1024-dimensional vector.

Although the ST dataset is inherently paired, it is constrained by the limited number of treatment groups and downstream labels. Additionally, obtaining ST datasets remains challenging and expen-

Table 2: A sub list of Chemical Structures in SMILES Notation for COMP Dataset

| SMILES Notation |
| --- |
| CC(C)(C#N)c1ccc(N2C(=O)OCc3cnc4ccc(-c5cnc6ccccc6c5)cc4c32)cc1 |
| CC(C)(C)c1ccc(C(=O)NCCC(=O)Nc2ccc3[nH]ncc3c2)cc1 |
| CC(C)(O)COc1ccc(S(=O)(=O)Nc2cccc3c(C#N)c[nH]c23)cc1 |
| CC(C)C1CCc2[nH]c(=O)c(C#N)cc2C1 |
| CC(C)Oc1ccc(-c2nc(-c3cccc4c3CCC4NCCC(=O)O)no2)cc1C#N |
| CC(C)c1nn(C)c2nc(C(F)F)nc(NCc3cccc(Oc4ccccn4)c3)c12 |
| CC(C)n1ncc2nccc(OCc3ccc(Br)cc3)c21 |
| CC1CCC(n2c3cnccc3c3cnc(Nc4ccc5c(n4)CCN(C(=O)CO)C5)nc32)CC1 |
| CC1Sc2ccc(C(=O)Nc3ccc(-c4cn5ccccc5n4)cc3)cc2NC1=O |
| CCCn1cnc2ncc(Nc3ccc4c(c3)CCC4)nc21 |
| CCOC(=O)c1c(C)nc2c(c1-c1ccc(Cl)cc1)C(=O)CC(C)(C)C2 |
| CCc1nc(S(=O)(=O)Nc2ccc(C)c3c(C#N)c[nH]c23)c[nH]1 |
| CCn1c(O)nc2cc(Cl)c(Cl)cc21 |
| CNC(=O)C(NC(=O)C(CC(C)C)C(O)C(=O)NO)C(C)(C)C |
| COc1cc(-c2cc3c(cn2)c(C2CC2)nn3C(C)C)ccn1 |
| COc1ccc(C(=O)Nc2ccc(-c3cn[nH]c3)c(C)c2)nc1 |
| COc1ccc(C2CNC(=O)C2)cc1OC1CCCC1 |
| COc1ccc(OC2CCN(c3cn[nH]c(=O)c3Cl)CC2)cc1 |
| COc1ccc2nc3c([N+](=O)[O-])ccc4c3c(c2c1)NN4CCCN(C)C |
| COc1nc(C)cnc1Nc1cc2c(cn1)c(C1CC1)nn2C(C)C |
| CS(=O)(=O)N1CCN(Cc2cc3nc(-c4cccc5[nH]ncc45)nc(N4CCOCC4)c3s2)CC1 |
| CS(=O)(=O)c1ccc(-c2nc(NCc3ccc4c(c3)OCO4)c3cn[nH]c3n2)cc1 |
| Cc1c(C(=O)Nc2ccccc2Cc2cccnc2)oc2ccc(Br)cc12 |
| ... |

sive, often confined to laboratory environments. This limits broader adoption within the biological community, especially for single-cell level experiments.

**A.3. Compound-Treated Single-Cell Dataset (COMP)**

To better understand multimodal learning for biological datasets, we curated two unpaired biological datasets at the single-cell level. The first dataset, referred to as the compound-treated single-cell dataset (COMP), originates from two separate biological experiments that studied the same cell type, human umbilical vein endothelial cells (HUVEC) (Baudin et al., 2007). Both experiments involved the same set of FDA-approved bioactive small-molecule compounds, similar to those used by Fay et al. (2023), applied at varying concentration levels (1, 2.5, and 10) in $\mu M$. In the first experiment, HUVEC cells were treated with compounds in multi-well plates and stained using the cell-painting protocol (Baudin et al., 2007), generating 6-channel fluorescent microscopy images ($X^1$). In the second experiment, cells were treated identically, but their RNA expression profiles were measured using bulk RNA sequencing (Ye et al., 2018) ($X^2$).

We post-processed each modality separately. The cell-painting images were cropped into $32 \times 32$ single-cell image crops, which were then embedded into a 786-dimensional vector using the *Phenom-1* model (Kraus et al., 2024) (pretrained encoder $\mathcal{F}_1$), a 300 million parameter pretrained foundation model designed for large scale analysis of microscopy images. For the sequencing data, we trained a scVI model on all sequences and embedded each sequence into a 256-dimensional vector ($\mathcal{F}_2$). A total of 78 small-molecule compounds were selected as treatments ($T$), with a sublist of their SMILES representations shown in Table 2.

To construct the COMP dataset, we randomly paired each sequence with approximately four image crops from the same treatment group, resulting in 36,562 image-sequence pairs. Each pair is annotated with a treatment group label ($t$) and a shared concentration level label ($Y$), serving as downstream task labels.

### A.4. Gene Knockout Single-cell Dataset (GKO)

Table 3: List of Targeted Knockout Genes

| Gene Name |
| --- |
| ATP6V1E1 |
| IL6R |
| CD33 |
| HPS1 |
| CD22 |
| ERGIC1 |
| CASP6 |
| ADORA2B |
| EIF2AK3 |
| CANX |
| ARF1 |
| ATP13A2 |
| BST1 |
| ATP1A3 |
| ABI3 |
| ALS2 |
| SYNJ1 |
| AKT1 |
| AKT2 |
| FOXRED1 |
| ACVRL1 |
| PSMA7 |
| ATP6V1F |
| EIF2B1 |

The second unpaired biological dataset we curated, referred to as the Gene Knockout Single-cell Dataset (GKO), follows a similar structure to the COMP dataset but focuses on CRISPR-mediated gene knockouts as treatments (Bock et al., 2022). The sequencing modality employs single-cell RNA sequencing (Dixit et al., 2016) to capture gene expression profiles. A visualization of the cell images is shown in Figure 7.

As with COMP, two separate experiments were conducted on the same HUVEC cell line. In the first experiment, HUVEC cells were treated with CRISPR-mediated knockouts targeting specific genes. After treatment, the cells were stained using the cell-painting protocol (Bray et al., 2016), producing 6-channel fluorescent microscopy images ($X^1$). In the second experiment, the same set of knockouts was applied, followed by single-cell RNA sequencing (Dixit et al., 2016) to generate RNA expression profiles ($X^2$).

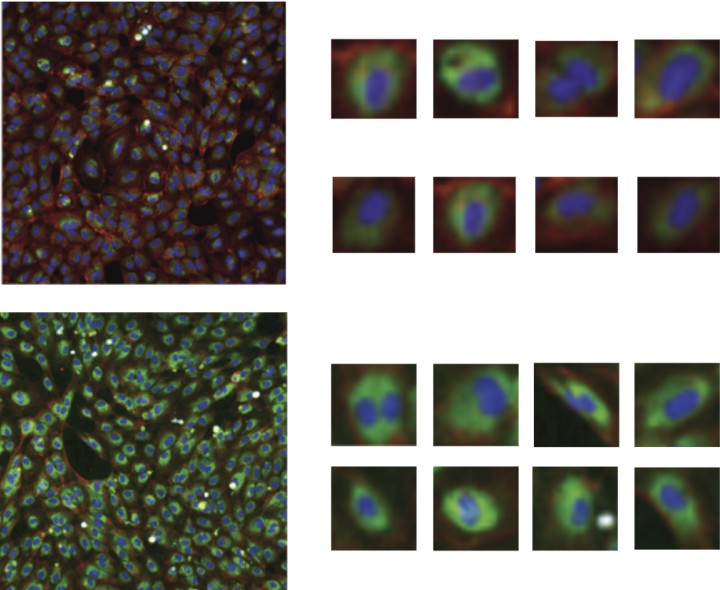

Figure 7: **Cell-painting images.** Both the COMP and GKO datasets utilize fluorescent microscopy images (left) obtained from biological experiments. Each image includes the locations of cell nuclei, enabling us to crop the images into smaller, single-cell images (right) for downstream processing.

Table 4: Dataset specifications. $N$: samples, $|T|$: treatments, $|Y|$: labels. Dimensions $|X^{1,2}|$ correspond to encoders $\mathcal{F}^{1,2}$.

| Dataset | $N$ | $|T|$ | $|Y|$ | $\mathcal{F}^1$ (Emb.) | $\mathcal{F}^2$ (Emb.) | $|X^1|$ | $|X^2|$ | Paired |
|---------|-----|-------|-------|------------------------|------------------------|---------|---------|--------|
| BIRFL | 5,502 | 7 | 3 | ViT(Dosovitskiy, 2020) | SFR(Lee et al., 2024) | 768 | 4096 | ✓ |
| ST | 57,363 | 4 | 2 | UNI(Chen et al., 2024) | scVI(Lopez et al., 2018) | 1024 | 1024 | ✓ |
| COMP | 36,562 | 78 | 3 | Phenom1(Kraus et al., 2024) | scVI(Lopez et al., 2018) | 768 | 256 | X |
| GKO | 59,011 | 25 | 5 | Phenom1(Kraus et al., 2024) | scVI(Lopez et al., 2018) | 1024 | 128 | X |

The image modality was post-processed into $32 \times 32$ single-cell image crops, which were then embedded into 1024-dimensional vectors using the Phenom-1 model (Kraus et al., 2024) ($\mathcal{F}_1$). For the single-cell sequencing data, we trained an scVI model (Lopez et al., 2018) ($\mathcal{F}_2$) on all sequences, embedding each sequence into a 128-dimensional vector. The treatment set $T$ included negative control + 25 targeted genes (listed in Table 3), and the downstream task labels ($Y$) were derived by discretizing the total gene counts into 5 levels, reflecting biological variability in gene expression profiles across treatments.

To construct the GKO dataset, we randomly paired single-cell images with single-cell sequences within the same treatment groups, resulting in a total of 59,011 image-sequence pairs.

## Appendix B. Experiment Details

This section outlines the experimental setup, detailing the learning objectives for each method and the training details for our experiments.

### B.1. Contrastive Learning Objectives

We provide the detailed learning objectives for each baseline method used in our experiments, as well as the objective introduced in our proposed IPIC method.

In the unpaired setting, a batch $\{(x_i^1, x_j^2)\}_{i,j=1}^B$ consists of samples from different modalities linked only by their shared treatment label ($t_i = t_j$), resulting in hidden vectors $\{(v_i^1, v_j^2)\}_{i,j=1}^B$. Given a unpaired batch of hidden vectors, the NCE objective becomes

$$\mathcal{L}_{NCE}^{i(1)} = -\log \frac{\mathbb{1}_{i=j} \exp(\mathrm{sim}(v_i^1, v_j^2)/\tau)}{\sum_{l=1}^B \mathbb{1}_{[l \neq i]} \exp(\mathrm{sim}(v_i^1, v_l^2)/\tau)}$$

$$\mathcal{L}_{NCE}^{j(2)} = -\log \frac{\mathbb{1}_{j=i} \exp(\mathrm{sim}(v_j^2, v_i^1)/\tau)}{\sum_{l=1}^B \mathbb{1}_{[l \neq i]} \exp(\mathrm{sim}(v_j^2, v_l^1)/\tau)}$$

$$\mathcal{L}_{NCE} = \frac{1}{2B} \sum_i^B (\mathcal{L}_{NCE}^{i(1)} + \mathcal{L}_{NCE}^{j(2)})$$

The full definition of SupCon objective is:

$$\mathcal{L}_{SupCon}^{i(1)} = -\sum_j^B \log \frac{\mathbb{1}_{[t_i=t_j]} \cdot \exp(\mathrm{sim}(v_i^1, v_j^2)/\tau)}{\sum_{l=1}^B \mathbb{1}_{[l \neq j]} \exp(\mathrm{sim}(v_i^1, v_l^2)/\tau)}$$

$$\mathcal{L}_{SupCon}^{j(2)} = -\sum_i^B \log \frac{\mathbb{1}_{[t_j=t_i]} \cdot \exp(\mathrm{sim}(v_j^2, v_i^1)/\tau)}{\sum_{l=1}^B \mathbb{1}_{[l \neq i]} \exp(\mathrm{sim}(v_j^2, v_l^2)/\tau)}$$

$$\mathcal{L}_{SupCon} = \frac{1}{2B} \sum_i^B (\mathcal{L}_{SupCon}^{i(1)} + \mathcal{L}_{SupCon}^{j(2)})$$

According to Wang et al. (2024), for our experiments we treat the infoCore objective as

$$\mathcal{L}_{InfoCore} = \frac{1}{2} \mathcal{L}_{SupCon} + \frac{1}{2} \mathcal{L}_{NCE}.$$

For an unpaired batch, the objective of WCL (Zheng et al., 2021) becomes:

$$\mathcal{L}_{WCL}^{i(1)} = -\sum_j^B \log \frac{\mathbb{1}_{[c_i^2=c_j^2]} \cdot \exp(\mathrm{sim}(v_i^1, v_j^2)/\tau)}{\sum_{l=1}^B \mathbb{1}_{[l \neq j]} \exp(\mathrm{sim}(v_i^1, v_l^2)/\tau)}$$

$$\mathcal{L}_{WCL}^{j(2)} = -\sum_i^B \log \frac{\mathbb{1}_{[c_j^1=c_i^1]} \cdot \exp(\mathrm{sim}(v_j^2, v_i^1)/\tau)}{\sum_{l=1}^B \mathbb{1}_{[l \neq i]} \exp(\mathrm{sim}(v_j^2, v_l^1)/\tau)}$$

$$\mathcal{L}_{WCL} = \frac{1}{2B} \sum_i^B (\mathcal{L}_{WCL}^{i(1)} + \mathcal{L}_{WCL}^{j(2)})$$

Here $c_i^\blacksquare$ and $c_j^\blacksquare$ denote the cluster label of the $i$ and $j$th index of modality $\blacksquare$. Thus, modality 1 objective utilize the modality 2's cluster labels as targets and vice versa for modality 2. We follow the same

implementation as Zheng et al. (2021) using connected components algorithms to find the cluster labels.

For XDC (Alwassel et al., 2020), where no publicly available implementation is available, we followed the description in the paper. Specifically, we used K-means as the clustering method. Instead of using an NCE-type objective, we directly employed the cross-modality labels as classification targets and calculated the cross-entropy loss between $v^1$ and $c^2$ or $v^2$ and $c^1$. We choose the number of clusters as the number of treatment for dataset for convenience

$$\mathcal{L}_{XDC} = -\frac{1}{2B} \sum_{i=1}^{B} \sum_{t=1}^{|T|} c_{i,t}^2 \log(v_i^1) - \frac{1}{2B} \sum_{j=1}^{B} \sum_{t=1}^{|T|} c_{j,t}^1 \log(v_j^2)$$

In all objectives above, $i$ indexes samples in modality 1, and $j$ corresponds to the $i$th sample in modality 2. Then $l$ is the denominator index for the sum of other similarities calculations.

For our one-time matching objective, after applying Algorithm 1, the dataset is pseudo-paired using the matching matrices for each treatment group. The indexing of modality 2 is updated to be indexed by $k$. Thus, the batch of hidden vectors becomes $\{(\mathbf{v}_i^1, \mathbf{v}_k^2)\}_{i,k=1}^{B}$, where $t_i = t_k$. Our one-time matching objective is then defined as:

$$\mathcal{L}_{1X-Matching}^{i(1)} = -\log \frac{\mathbb{1}_{i=k} \exp(\mathrm{sim}(\mathbf{v}_i^1, \mathbf{v}_k^2)/\tau)}{\sum_{l=1}^{B} \mathbb{1}_{[l \neq i]} \exp(\mathrm{sim}(\mathbf{v}_i^1, \mathbf{v}_l^2)/\tau)}$$

$$\mathcal{L}_{1X-Matching}^{k(2)} = -\log \frac{\mathbb{1}_{k=i} \exp(\mathrm{sim}(\mathbf{v}_k^2, \mathbf{v}_i^1)/\tau)}{\sum_{l=1}^{B} \mathbb{1}_{[l \neq i]} \exp(\mathrm{sim}(\mathbf{v}_k^2, \mathbf{v}_l^1)/\tau)}$$

$$\mathcal{L}_{1X-Matching} = \frac{1}{2B} \sum_{i}^{B} (\mathcal{L}_{1X-Matching}^{i(1)} + \mathcal{L}_{1X-Matching}^{k(2)})$$

As our $\mathcal{L}_{intra}$ and $\mathcal{L}_{inter}$ introduced in our IPIC method also compute objectives on pseudo-paired batches, the full objective of $\mathcal{L}_{intra}$ becomes:

$$\mathcal{L}_{intra}^{i(1)} = -\sum_{k}^{B} \log \frac{\mathbb{1}_{[t_i=t_k]} \cdot m_{i,k}^1 \cdot \exp(\mathrm{sim}(\mathbf{v}_i^1, \mathbf{v}_k^2)/\tau)}{\sum_{l=1}^{B} \mathbb{1}_{[l \neq k]} \exp(\mathrm{sim}(\mathbf{v}_i^1, \mathbf{v}_l^2)/\tau)},$$

$$\mathcal{L}_{intra}^{k(2)} = -\sum_{i}^{B} \log \frac{\mathbb{1}_{[t_k=t_i]} \cdot m_{k,i}^2 \cdot \exp(\mathrm{sim}(\mathbf{v}_k^2, \mathbf{v}_i^1)/\tau)}{\sum_{l=1}^{B} \mathbb{1}_{[l \neq i]} \exp(\mathrm{sim}(\mathbf{v}_k^2, \mathbf{v}_l^1)/\tau)},$$

$$\mathcal{L}_{intra} = \frac{1}{2B} \sum_{i,k}^{B} (\mathcal{L}_{intra}^{i(1)} + \mathcal{L}_{intra}^{k(2)}),$$

The full objective of $\mathcal{L}_{inter}$ is

$$\mathcal{L}_{inter}^{i(1)} = -\sum_{o}^{B} \log \frac{\mathbb{1}_{[i=k]} \cdot \mathbb{1}_{[c_k^2=c_o^2]} \cdot \exp(\mathrm{sim}(\mathbf{u}_i^1, \mathbf{u}_o^2)/\tau)}{\sum_{l=1}^{B} \mathbb{1}_{[l \neq k]} \exp(\mathrm{sim}(\mathbf{u}_i^1, \mathbf{u}_l^2)/\tau)},$$

$$\mathcal{L}_{inter}^{k(2)} = -\sum_{o}^{B} \log \frac{\mathbb{1}_{[k=i]} \cdot \mathbb{1}_{[c_i^1=c_o^1]} \cdot \exp(\mathrm{sim}(\mathbf{u}_k^2, \mathbf{u}_o^1)/\tau)}{\sum_{l=1}^{B} \mathbb{1}_{[l \neq i]} \exp(\mathrm{sim}(\mathbf{u}_k^2, \mathbf{u}_l^1)/\tau)},$$

$$\mathcal{L}_{inter} = \frac{1}{2B} \sum_{i,k}^{B} (\mathcal{L}_{inter}^{i(1)} + \mathcal{L}_{inter}^{k(2)}),$$

Here, $o$ denotes the set of indices for all hidden vectors in the cross-modality that share the same cluster label as the pseudo-paired sample in the cross-modality.

### B.2. Training Details

In this section, we describe the training and experimental details, including architecture choices and the hyperparameters used for our experiments.

To perform matching, we first train a propensity score predictor $\psi$ for each modality across all four datasets. Since $X^1$ and $X^2$ for each dataset are represented as vectors, we train two separate MLPs, one for each modality. Each MLP has two hidden layers, with the input dimensions corresponding to $|X^1|$ and $|X^2|$, respectively, and the output dimension equal to $|T|$, as shown in Table 4. Each $\psi$ is trained using only the vectors of a single modality and their corresponding treatment label $t$.

Table 5: Model Input and Output Dimensions for Each Dataset

| Model/Map | BIRFL | ST | COMP | GKO |
|---|---|---|---|---|
| $\psi^1$ | $\mathbb{R}^{768} \to \mathbb{R}^{7}$ | $\mathbb{R}^{1024} \to \mathbb{R}^{4}$ | $\mathbb{R}^{768} \to \mathbb{R}^{78}$ | $\mathbb{R}^{1024} \to \mathbb{R}^{25}$ |
| $\psi^2$ | $\mathbb{R}^{4096} \to \mathbb{R}^{7}$ | $\mathbb{R}^{1024} \to \mathbb{R}^{4}$ | $\mathbb{R}^{256} \to \mathbb{R}^{78}$ | $\mathbb{R}^{128} \to \mathbb{R}^{25}$ |
| $\phi^1$ | $\mathbb{R}^{768} \to \mathbb{R}^{512}$ | $\mathbb{R}^{1024} \to \mathbb{R}^{512}$ | $\mathbb{R}^{768} \to \mathbb{R}^{512}$ | $\mathbb{R}^{1024} \to \mathbb{R}^{512}$ |
| $\phi^2$ | $\mathbb{R}^{4096} \to \mathbb{R}^{512}$ | $\mathbb{R}^{1024} \to \mathbb{R}^{512}$ | $\mathbb{R}^{256} \to \mathbb{R}^{512}$ | $\mathbb{R}^{128} \to \mathbb{R}^{512}$ |
| $f^1, f^2$ | $\mathbb{R}^{512} \to \mathbb{R}^{512}$ | $\mathbb{R}^{512} \to \mathbb{R}^{512}$ | $\mathbb{R}^{512} \to \mathbb{R}^{512}$ | $\mathbb{R}^{512} \to \mathbb{R}^{512}$ |
| $g^1, g^2$ | $\mathbb{R}^{512} \to \mathbb{R}^{512}$ | $\mathbb{R}^{512} \to \mathbb{R}^{512}$ | $\mathbb{R}^{512} \to \mathbb{R}^{512}$ | $\mathbb{R}^{512} \to \mathbb{R}^{512}$ |

For all four datasets, we adopt the CLIP model architecture as implemented by the Hugging Face platform with the default Vision and Language configuration. However, we customize the model by excluding the encoder parts for both the image and text modalities, starting directly with the hidden and projection layers. For each dataset, the input modality dimensions are linearly projected into the CLIP model's hidden vector dimension. Both $\phi^1$ and $\phi^2$ produce 512-dimensional vectors, which serve as inputs to the downstream projection heads. Each projection head $f$ and $g$ is implemented as a three-layer MLP comprising hidden linear layers, ReLU activations, and batch normalization. The detailed input and output dimension of models with trainable parameters for each dataset are shown in Table 5.

Table 6: Hyperparameter Choices for Each Dataset

| Hyperparameter | BIRFL | ST | COMP | GKO |
|---|---|---|---|---|
| Learning Rate | $5 \times 10^{-7}$ | $1 \times 10^{-4}$ | $1 \times 10^{-6}$ | $1 \times 10^{-5}$ |
| Batch Size | 512 | 4096 | 4096 | 4096 |
| Epochs | 50 | 200 | 100 | 100 |

For each dataset, we split the training and testing datasets in an 80-20 split fashion, with the split performed randomly and controlled by a random seed. For each dataset, we ran every hyperparameter setting using 8 random seeds. We experimented with learning rates { 1e-3, 1e-4, 1e-5, 1e-6, 5e-7 }, number of epochs {20, 50, 100, 200}, and batch sizes {128, 256, 512, 1024, 2048, 4096}. For the results reported, we selected the combination of learning rate, number of epochs, and batch size that achieved the highest average performance across all methods. The final results were reported as the mean and standard deviation over the 8 random seeds. The detailed hyperparameter selections for

Table 7: Test Set Results on Our Four Curated Datasets with Treatment and Downstream Task Label Predictions (Mean $\pm$ Std). **Bold** indicates best performance.

| | BIRFL | | ST | | COMP | | GKO | |
|---|---|---|---|---|---|---|---|---|
| Method | Treatment | Task | Treatment | Task | Treatment | Task | Treatment | Task |
| *Reference Baselines* | | | | | | | | |
| Raw Features ($x^1||x^2$) | $77.0 \pm 0.4$ | $55.4 \pm 1.2$ | $77.7 \pm 0.3$ | $63.6 \pm 0.5$ | $74.3 \pm 1.1$ | $58.6 \pm 1.4$ | $19.9 \pm 0.2$ | $28.9 \pm 0.6$ |
| InfoNCE-paired (Oracle) | $81.4 \pm 1.3$ | $97.2 \pm 1.1$ | $98.5 \pm 0.1$ | $80.9 \pm 0.2$ | - | - | - | - |
| FactorCL-SSL | $80.7 \pm 1.5$ | $64.5 \pm 2.8$ | $92.1 \pm 0.4$ | $67.3 \pm 0.6$ | $90.9 \pm 2.1$ | $72.5 \pm 1.9$ | $25.1 \pm 1.1$ | $37.9 \pm 1.3$ |
| FactorCL-Sup | $81.0 \pm 1.2$ | $66.5 \pm 2.5$ | $95.7 \pm 0.3$ | $68.9 \pm 0.5$ | $91.6 \pm 2.4$ | $73.4 \pm 2.1$ | $30.7 \pm 1.5$ | $35.2 \pm 1.4$ |
| InfoNCE-unpaired(Chen et al., 2020) | $81.4 \pm 1.4$ | $64.7 \pm 3.6$ | $97.0 \pm 0.2$ | $68.4 \pm 0.4$ | $95.6 \pm 3.6$ | $72.9 \pm 2.4$ | $22.6 \pm 1.6$ | $36.9 \pm 1.0$ |
| SupCon(Khosla et al., 2020) | $83.0 \pm 1.0$ | $65.3 \pm 3.4$ | $\mathbf{99.4 \pm 0.2}$ | $68.9 \pm 0.5$ | $96.7 \pm 2.5$ | $73.4 \pm 2.4$ | $\mathbf{34.1 \pm 1.8}$ | $33.7 \pm 1.5$ |
| InfoCore(Wang et al., 2024) | $82.1 \pm 3.5$ | $65.2 \pm 0.9$ | $98.8 \pm 0.2$ | $68.4 \pm 0.4$ | $96.3 \pm 3.0$ | $73.6 \pm 2.6$ | $27.8 \pm 1.6$ | $34.9 \pm 1.6$ |
| WCL(Zheng et al., 2021) | $81.1 \pm 0.9$ | $67.4 \pm 2.3$ | $96.6 \pm 0.2$ | $68.8 \pm 0.4$ | $96.4 \pm 2.9$ | $75.6 \pm 2.2$ | $23.2 \pm 1.6$ | $36.8 \pm 0.8$ |
| XDC(Alwassel et al., 2020) | $80.8 \pm 1.0$ | $67.2 \pm 3.3$ | $94.7 \pm 0.3$ | $68.4 \pm 0.4$ | $96.2 \pm 2.8$ | $77.6 \pm 2.2$ | $21.8 \pm 1.4$ | $37.3 \pm 0.9$ |
| *Matching Strategy Ablation* | | | | | | | | |
| 1X-Matching (SNN) | $74.2 \pm 1.8$ | $65.2 \pm 2.1$ | $93.9 \pm 0.5$ | $68.2 \pm 0.6$ | $91.1 \pm 3.1$ | $74.2 \pm 2.5$ | $19.7 \pm 1.9$ | $37.5 \pm 1.2$ |
| 1X-Matching (OT) | $81.5 \pm 0.5$ | $66.7 \pm 2.5$ | $95.9 \pm 0.2$ | $68.6 \pm 0.5$ | $95.7 \pm 3.4$ | $73.0 \pm 2.2$ | $23.6 \pm 1.5$ | $36.6 \pm 1.0$ |
| Ours (intra-SNN) | $73.9 \pm 0.8$ | $69.4 \pm 1.2$ | $96.5 \pm 0.3$ | $68.0 \pm 0.5$ | $92.4 \pm 1.5$ | $76.1 \pm 1.8$ | $20.7 \pm 1.1$ | $41.3 \pm 0.9$ |
| Ours (intra-OT) | $83.4 \pm 0.3$ | $69.7 \pm 0.3$ | $97.8 \pm 0.3$ | $70.4 \pm 0.4$ | $97.4 \pm 0.3$ | $78.5 \pm 0.9$ | $26.0 \pm 0.4$ | $42.8 \pm 0.4$ |
| *Clustering Ablation (IPIC)* | | | | | | | | |
| Ours (intra+inter, $K = 0.25|T|$) | - | - | - | - | $97.3 \pm 0.6$ | $80.1 \pm 1.0$ | $25.6 \pm 0.5$ | $39.4 \pm 0.7$ |
| Ours (intra+inter, $K = 0.5|T|$) | $82.3 \pm 0.6$ | $66.4 \pm 1.5$ | $96.7 \pm 0.3$ | $70.0 \pm 0.4$ | $97.4 \pm 0.5$ | $\mathbf{80.3 \pm 0.8}$ | $25.4 \pm 0.6$ | $43.0 \pm 0.5$ |
| **Ours (intra+inter, $K = |T|$)** | $\mathbf{83.5 \pm 0.2}$ | $\mathbf{71.5 \pm 0.5}$ | $97.7 \pm 0.1$ | $\mathbf{70.7 \pm 0.2}$ | $\mathbf{97.8 \pm 0.2}$ | $80.2 \pm 0.7$ | $26.4 \pm 0.5$ | $\mathbf{43.3 \pm 0.2}$ |
| Ours (intra+inter, $K = 2|T|$) | $83.5 \pm 0.3$ | $69.3 \pm 0.9$ | $97.0 \pm 0.2$ | $70.1 \pm 0.3$ | $97.3 \pm 0.4$ | $79.7 \pm 0.8$ | $23.6 \pm 0.5$ | $42.0 \pm 0.4$ |
| Ours (intra+inter, $K = 4|T|$) | $82.8 \pm 0.5$ | $69.2 \pm 1.1$ | $96.9 \pm 0.2$ | $70.0 \pm 0.4$ | $97.2 \pm 0.5$ | $79.0 \pm 0.9$ | $23.7 \pm 0.6$ | $42.2 \pm 0.3$ |

each dataset are shown in Table 6. These hyperparameters were chosen to ensure consistency and comparability across experiments.

## Appendix C. Extra Experiments and Results

This section provides a comprehensive evaluation of the IPIC embeddings in the context of un-paired multimodal learning to show its effectiveness at capturing biological knowledge. We focus on metrics such as *Perturbation Consistency*, *iLISI*, and *Structural Integrity*, adapted to unpaired data, where samples across modalities share treatment labels but lack direct pairwise correspondence.

### C.1. Full Results

We present the full results, including both the mean and standard deviation, in Table 7. These results supplement the mean-only results reported in Table 1 in the main text, providing a more comprehensive view of the performance across methods and datasets.

For each experiment conducted, we first use the learned model to embed the training dataset. The embeddings from both modalities are then concatenated to form a multi-modality representation. Using these concatenated vectors, we train a logistic regression model, which is subsequently used to predict the labels on the concatenated test set embeddings. This process provides the accuracy for the zero-shot performance of the test set embeddings.

### C.2. Single-Modality vs. Multi-Modality

In order to compare whether multi-modality embedding outperforms single-modality embedding, for each method and each dataset, we use the learned encoders $\phi^1$ and $\phi^2$ to obtain embeddings for each modality. These embeddings are then used to perform zero-shot prediction tasks on the downstream task label $Y$. From the results shown in Table 8, we observe that, with the exception of the image modality outperforming the concatenated multi-modality embedding in the ST dataset, all other datasets demonstrate that multi-modality training is more beneficial than single-modality

Table 8: Comparison of Performance Across Four Datasets with Single- and Multi-Modal Averages

| | BIRFL | | | | ST | | | | COMP | | | | GKO | | | |
|---|---|---|---|---|---|---|---|---|---|---|---|---|---|---|---|---|
| **Method** | mod1 | mod2 | avg | multimod | mod1 | mod2 | avg | multimod | mod1 | mod2 | avg | multimod | mod1 | mod2 | avg | multimod |
| InfoNCE | 56.2 | 33.5 | 44.9 | 64.7 | 69.9 | 60.7 | 65.3 | 68.4 | 69.8 | 38.9 | 54.8 | 72.9 | 27.1 | 32.6 | 29.9 | 36.9 |
| SupCon | 58.5 | 36.1 | 47.3 | 65.3 | 70.4 | 61.2 | 65.8 | 68.9 | 70.4 | 38.5 | 54.5 | 73.4 | 27.2 | 31.7 | 29.5 | 33.7 |
| 1X-Matching | 60.5 | 37.7 | 49.1 | 66.7 | 69.6 | 60.7 | 65.2 | 68.6 | 70.4 | 37.9 | 54.2 | 73.0 | 27.2 | 33.4 | 30.3 | 36.6 |
| WCL | 63.9 | 37.6 | 50.8 | 67.4 | 69.4 | 61.0 | 65.2 | 68.8 | 72.9 | 38.8 | 55.9 | 75.6 | 23.8 | 33.7 | 28.8 | 36.8 |
| Ours | 64.6 | 39.5 | 52.1 | 71.5 | 72.2 | 62.1 | 67.2 | 70.7 | 74.5 | 39.4 | 57.0 | 80.2 | 27.9 | 41.2 | 34.6 | 43.3 |

embeddings. Multi-modality learning enhances both data modalities, as the average performance of single-modality embeddings for downstream tasks is consistently lower than that of concatenated multi-modality embeddings. Specifically, the accuracy drops by 4% to 26% when using single-modality embeddings. Additionally, Table 8 highlights a trend where our proposed IPIC method consistently improves the quality of each single-modality embedding compared to baseline methods, further demonstrating its effectiveness in multi-modality learning.

### C.3. Perturbation Consistency

The *Perturbation Consistency* metric evaluates how consistently a model captures the effects of treatments across samples and batches. For unpaired data, this involves assessing similarities within and across treatment groups for embeddings from each modality. The metric is computed as follows:

**1. Average Within-Group Similarity:**

Given embeddings $\mathbf{v}_i$ for samples in treatment group $T_k$, the cosine similarity between embeddings is computed as:

$$\cos(\mathbf{v}_i, \mathbf{v}_j) = \frac{\mathbf{v}_i \cdot \mathbf{v}_j}{\|\mathbf{v}_i\| \|\mathbf{v}_j\|}. \tag{11}$$

The average similarity for group $T_k$ is:

$$\text{AvgSim}_{T_k} = \frac{1}{|T_k|^2} \sum_{i,j \in T_k} \cos(\mathbf{v}_i, \mathbf{v}_j). \tag{12}$$

**2. Contrast Between Groups:** For embeddings from different treatment groups $T_k$ and $T_l$ ($k \neq l$), the inter-group similarity is:

$$\text{AvgSim}_{T_k, T_l} = \frac{1}{|T_k||T_l|} \sum_{i \in T_k, j \in T_l} \cos(\mathbf{v}_i, \mathbf{v}_j) \tag{13}$$

**3. Perturbation Consistency Score:** The final score measures the separation of within-group similarities from across-group similarities:

$$\text{Perturbation Consistency} = \frac{\sum_k \text{AvgSim}_{T_k}}{\sum_{k,l,k \neq l} \text{AvgSim}_{T_k, T_l}} \tag{14}$$

For simplicity, we concatenated embeddings from both modalities for each "paired sample" to compute the Perturbation Consistency score.

Higher scores indicate better preservation of treatment effects within groups. From the results shown in Table 9, we observe that the Perturbation Consistency score computed from the original input vectors $X^1$ and $X^2$ is lowest across the ST, COMP, and GKO datasets. For the BIRFL dataset, methods such as InfoNCE, one-time matching, and WCL slightly decrease the perturbation consistency of the embeddings. However, our IPIC method consistently outperforms these methods,

Table 9: Perturbation Consistency

| Method | BIRFL | ST | COMP | GKO |
|---|---|---|---|---|
| Original Input | 0.178 | 0.337 | 0.085 | 0.042 |
| InfoNCE | 0.171 | 0.371 | 0.086 | 0.083 |
| SupCon | 0.278 | 0.438 | 0.109 | 0.120 |
| WCL | 0.176 | 0.389 | 0.103 | 0.043 |
| 1X-Matching | 0.172 | 0.372 | 0.087 | 0.086 |
| IPIC (Ours) | 0.180 | 0.433 | 0.114 | 0.129 |

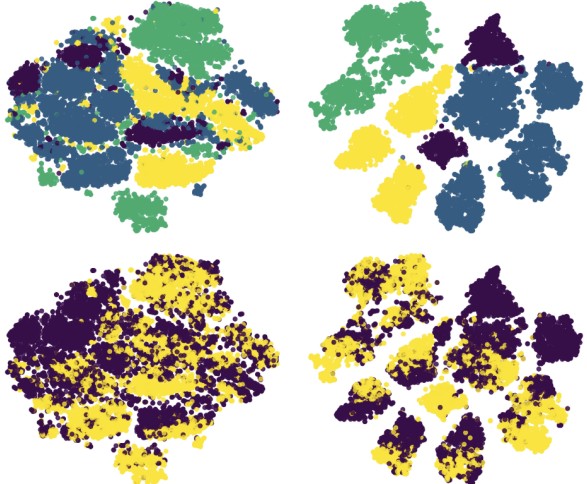

Figure 8: *t*-SNE visualization of embeddings. The plots illustrate separability by treatment class (*T*) (top) and downstream tasks labels (*Y*) (bottom) for pretraining embeddings (left) and IPIC embeddings (right).

including the original input vectors, and achieves better performance than SupCon, which directly uses treatment as labels during training. These results demonstrate that our IPIC method successfully preserves treatment effects within each treatment group, further validating its effectiveness.

### C.4. Data Integration and Batch Effect Reduction

In biological experiments, data are often collected in distinct batches (groups of samples processed under varying conditions or at different times). These batch-specific variations, known as *batch effects*, can obscure true biological signals and hinder accurate perturbation analysis. Effective batch integration is essential to ensure that comparisons in the latent space reflect genuine biological differences rather than technical artifacts.

The *Integration Local Inverse Simpson's Index (iLISI)* measures how well a model mitigates batch effects by evaluating the diversity of batch labels within the neighborhood of each sample in the embedding space (Korsunsky et al., 2019). A high iLISI score signifies successful batch mixing, enabling the latent space to highlight biological variation while minimizing technical noise. The iLISI computation follows these steps:

Table 10: iLISI Scores for Batch Integration

| Model | GKO (Batch ID) | ST (Slide ID) |
|---|---|---|
| IPIC ($\phi_1 \| \phi_2$) | 1.82 | 1.67 |
| $\mathcal{F}_1$ (Original Encoder) | 1.52 | 1.12 |
| $\mathcal{F}_2$ (Original Encoder) | 1.66 | 1.1 |
| $\phi_1$ | 1.78 | 1.66 |
| $\phi_2$ | 1.79 | 1.35 |

**1. Neighborhood Probabilities:** For each sample $i$, the conditional probability $p_{ic}$ of its neighbors belonging to batch $c$ is computed as:

$$p_{ic} = \sum_{\substack{j \in \mathcal{N}_i \\ l_j = c}} p_{ij},$$

where $\mathcal{N}_i$ is the set of $k$-nearest neighbors of sample $i$, $l_j$ is the batch label of neighbor $j$, and $p_{ij}$ is the probability defined as:

$$p_{ij} = \frac{\exp(-\beta_i d_{ij})}{\sum_{l \in \mathcal{N}_i} \exp(-\beta_i d_{il})}.$$

Here, $\beta_i$ is a scaling parameter ensuring that the entropy of $P_i = \{p_{ij}\}$ matches $\log(k)$.

**2. Inverse Simpson's Index:** For each sample $i$, the diversity of batch labels in its neighborhood is measured using the Inverse Simpson's Index:

$$\text{ISI}_i = \left( \sum_{c \in C} p_{ic}^2 \right)^{-1},$$

where $C$ is the set of all batch labels.

**3. Final iLISI Score:** The overall iLISI score is the mean ISI across all samples:

$$\text{iLISI} = \frac{1}{n} \sum_{i=1}^{n} \text{ISI}_i.$$

A higher iLISI score indicates better batch integration, as samples are more evenly distributed across all batches within their neighborhoods.

We evaluated the iLISI scores for embeddings generated by IPIC (concatenated embeddings) and compared them to baseline models, which include the original modality encoders with their respective pretraining schemes. The comparison focuses on batch integration across two datasets: GKO (with batch ID categories) and ST (with slide ID categories).

Table 10 presents the iLISI scores, providing a quantitative comparison of integration performance between IPIC and baseline models (individual original encoders $\mathcal{F}$ and learned encoders $\phi$). IPIC achieved consistently higher iLISI scores than the original encoders, demonstrating superior batch effect reduction and data integration. Notably, the learned encoders ($\phi$) for each modality also showed significant improvement over their respective pretrained embeddings.

These results emphasize IPIC's ability to leverage multimodal embeddings effectively, reducing batch effects and integrating samples into a unified representation space more robustly than the baseline approaches.

## C.5. Separability of Embeddings

The ability of embeddings to separate meaningful biological signals from batch effects or other confounding factors is a critical property for robust representation learning. We evaluated the separability of embeddings generated by IPIC on ST dataset using $t$-SNE visualizations, focusing on metadata categories such as treatment class ($T$) and downstream task labels ($Y$). This visualization allows for a qualitative assessment of how well the embeddings group similar treatments while also distinguish orthogonal downstream task labels. Figure 8 shows $t$-SNE plots comparing pre- and post-training embeddings for both modalities. Pretraining embeddings exhibited significant overlap between distinct treatment classes, often clustering according to batch effects. In contrast, post-training embeddings generated by IPIC demonstrated clear separation of treatment classes and better distinction between downstream task labels.

The $t$-SNE visualization highlights the enhanced ability of IPIC to separate embeddings by treatment class while reducing the influence of batch effects by separating the downstream task labels clearly. This improvement indicates that IPIC embeddings better capture biologically meaningful signals compared to baseline models. However, further quantitative evaluations, such as classification accuracy or silhouette scores, could provide additional insights into the robustness of these embeddings.

