# OpenReview forum: "Unpaired Multimodal Learning for Biological Datasets"
_MIDL.io/2026/Conference — MIDL 2026 Poster_

### Official Review · Reviewer_iNnK · 2026-01-08

**Confidence:** 3
**Preliminary Rating:** 5
**Final Rating:** 5

**Summary:**

The authors have proposed to develop a general multimodal representation learning framework that works without paired samples, specifically targeting biological datasets where paired multimodal measurements are often unavailable due to not collected at the same time or because of destructive measurement processes (e.g., sequencing vs. imaging). In this challenging setting, the authors aim to learn biologically meaningful, aligned embeddings using only shared treatment-group labels, rather than exact sample-level pairing. The authors have conducted experiments on four different biological datasets to prove their proposed approach of IntraPair InterCluster (IPIC).

**Strengths:**

The paper addressed an important issue in multimodal medical datasets, where there is often a scenario when samples modalities are not paired and matching, thus making the learning more challenging. In this case, several new methods need to be adopted such as modality alignment using unpaired samples to effectively have a good learning strategy. I think the problem targeted by the authors is extremely relevant and essential for today's datasets in healthcare domain. Below are the main strengths of this work:

1. Strong motivation grounded in real world constraints - The paper tackles a core limitation of multimodal learning in biology - the impracticality of paired data. This is a realistic and important problem that is often overllooked in multimodal AI works, and the authors have articulated it clearly and convincingly. The paper motivation is well-written.

2. Novel and well-designed methodological contribution - The combination of intra-treatment matching and inter-treatment clustering (IPIC) is conceptually interesting and well justified for this problem. IPIC goes beyond naive weak supervision by explicitly modeling both population-level similarity and cross-treatment biological overlap.

3. Careful positioning relative to prior work - I really like how the authors have demonstrated the previous similar and existing approaches where they have inspired from. The writing shows a deep understanding of existing approaches (InfoNCE, SupCon, XDC, WCL, OT-based matching) and clearly explains why these methods are insufficient or brittle in unpaired biological settings, and how IPIC then demonstrates to solve the problem.

4. Comprehensive experimental evaluation - Experiments span four diverse datasets, including both naturally paired and genuinely unpaired biological data. More importantly, the inclusion of both treatment prediction and orthogonal downstream tasks is strong, as it avoids over-optimizing for treatment labels alone.

5. Extensive ablation and diagnostic analyses - The authors have conducted extensive ablations on matching strategies, clustering granularity, and objective components. I really liked all the important details in the appendix section as well. Additional metrics such as perturbation consistency, iLISI, and t-SNE visualizations provide meaningful biological validation beyond accuracy.

6. Use of strong pretrained foundational encoders - The authors have also leveraged recent foundation models (UNI, Phenom-1, scVI, ViT) ensuring that improvements stem from the learning framework, not weak feature extraction. This strengthens their validity of the conclusions.

**Weaknesses:**

There are certain questions that I have in mind which require clarification:

1. Heavy reliance on treatment labels as a supervisory signal - While treatment labels are realistically available in many biological studies, the proposed approach assumes that treatment information is both accurate and sufficiently informative. However, there can be scenarios when noisy, heterogeneous, or poorly defined treatments are present and I want to know whether the authors tried to test the performance of IPIC in any of those cases? Is there a simulation of such cases possible if they are not available readily? Moreover, I think in the future the authors should also explore when the treatment labels are missing.

2. Assumptions underlying propensity score matching - The proposed intra-treatment matching relies on assumptions basically borrowed from causal inference for instance the identical conditional treatment distributions is assumed across modalities. While this is reasonable, these assumptions may not hold true in all biological settings, especially when modalities capture fundamentally different aspects of cellular state. Have the authors tried to examine when such an assumption is not valid? This can be a future work as well.

3. Limited analysis of failure modes - The authors have described the cases where IPIC succeeds, but provides little insight into when or why it might fail (e.g., treatments with no measurable effect in one modality, highly imbalanced treatment groups, or strong batch–treatment confounding). Could the authors please describe more on the failure cases as well and why it may fail?

4. Computational complexity and scalability discussion is missing - The proposed IPIC is a technically sound framework and involves iterative optimal transport, clustering, and multiple projection heads. While feasible to implement in the presented experiments, the scalability to much larger treatment spaces or higher-dimensional modalities is not discussed in detail. Will the authors face any scaling issues for their approach?

**Detailed Comments:**

Some more points to further improve the paper:

1. Analyse robustness to noisy or incomplete treatment labels - A small experiment or sensitivity analysis where treatment labels are perturbed, merged, or partially removed would further strengthen confidence in the method’s applicability to real-world biological studies. I suggest the authors to add this if possible in the final version.

2. Clarify the limits of the matching assumptions - A more explicit discussion (or empirical test) of scenarios where propensity-score–based matching may fail would help readers understand when IPIC should or should not be applied.

3. Explore end-to-end or partially end-to-end variants - While not required for this paper version, discussing how IPIC could be extended to jointly fine-tune encoders (or) why this may be undesirable in biology, would broaden the method’s scope as future work.

4. Provide more intuition on inter-treatment clustering behavior - Additional qualitative examples showing which treatments are clustered together (and why) could further support the biological interpretability of the inter-treatment objective.

5. Discuss scalability and practical deployment in more detail- Including runtime or memory complexity estimates of IPIC and also a brief guidance on how to scale IPIC to thousands of treatments or modalities, would increase practical relevance.

**Justification Of Final Rating:**

I thank the authors for clarifying my comments and their promise to include the additional assumption analysis in their future work.

It's a strong work as mentioned earlier and will be of high interest to the entire MIDL community.

**Justification Of The Preliminary Rating:**

This is a strong and an important paper that addresses an underexplored problem in multimodal biological learning in healthcare domain. The proposed IPIC framework is methodologically novel, biologically well-motivated, and empirically validated across four diverse datasets. While some assumptions and practical limitations remain (and I suggest can be clarified in the final version), they are reasonable for the scope of the current work and are clearly outweighed by the contributions. The applicability of this work will be highly interesting to the whole community of MIDL.

**Questions To Address In The Rebuttal:**

Please refer to the Weakness and Detailed Comments section.

---

> ### Author Response · Authors · 2026-01-24
> **Thank you for your review**
>
> We thank the reviewer for their strong endorsement and for highlighting the relevance of our work to the healthcare domain. We are gratified that you found our motivation grounded in real-world constraints, our methodology novel and well-justified, and our experimental evaluation comprehensive.
> > W1 W2 C1 treatment label reliance and matching assumptions
>
> We agree that the causal assumption (treating labels as proxies for latent alignment) is a strong starting assumption that may not hold perfectly in every biological setting. However, we designed IPIC specifically to be resilient to this; the inter-treatment clustering component serves to mitigate the cost of local assumption violations by capturing broader, cross-group structural similarities. While we believe this makes the method robust to moderate noise, we agree that a dedicated study on severe assumption violations (e.g., missing labels) is an important direction for future work.
> > W3, C2 failure modes
>
> Fundamentally, IPIC operates under the assumption that the two modalities share latent information that can be learned contrastively. Therefore, a primary failure mode would be scenarios where modalities are orthogonal or do not biologically complement each other; in such cases, no contrastive method would succeed. Conversely, if data were available in a perfectly paired format, standard InfoNCE would suffice, rendering IPIC's matching mechanisms unnecessary (though our experiments on BIRFL/ST show IPIC remains effective even in these derived settings).
>
> > W4 computational complexity and scalability
>
> We acknowledge the need for a more detailed discussion on scalability. We are currently finalizing a formal runtime and complexity analysis of the iterative optimal transport and clustering steps, which will be included as a dedicated table in the appendix of the camera-ready version.

---

### Official Review · Reviewer_Me1Q · 2026-01-11

**Confidence:** 3
**Preliminary Rating:** 5

**Summary:**

The authors present IntraPair InterCluster (IPIC), a new contrastive representation learning method specifically tailored for unpaired biological data. IPIC overcomes the lack of paired samples by combining intra-treatment matching with inter-treatment clustering, effectively using treatment labels as weak supervision. The study validates this approach on multiple benchmarks, showing significant improvements in embedding quality compared to current state-of-the-art methods.

**Strengths:**

The study is supported by comprehensive empirical validation across four diverse curated datasets, covering phenomics, transcriptomics, and spatial transcriptomics contexts. The extensive experimental results demonstrate that the proposed method consistently outperforms relevant baselines, validating its effectiveness in leveraging independently collected single-modality datasets for multimodal contrastive pre-training.

**Weaknesses:**

1. The proposed IPIC framework relies heavily on the InfoNCE loss function, which necessitates large batch sizes (e.g., $N=4,096$ as used in the experiments 1) to maintain a sufficient pool of negative samples for effective contrastive learning. However, this requirement presents a significant barrier for applicability to high-dimensional medical imaging (e.g., 3D MRI or CT), where GPU memory constraints typically limit training to very small batch sizes (e.g., 2–16). Consequently, the method's claim of generality across 'biological datasets' is limited, as it does not address how the contrastive mechanism can function effectively under the memory constraints inherent to 3D imaging modalities.

2. The evaluation does not demonstrate that the method outperforms established state-of-the-art (SOTA) algorithms specifically designed for unpaired multi-omics integration, such as 'Jointly aligning cells and genomic features of single-cell multi-omics data with co-optimal transport' (SCOT). While the authors compare against a self-constructed '1X-Matching (OT)' baseline, they do not benchmark against specialized solvers like SCOT that utilize advanced co-optimal transport strategies. Without this direct comparison, it is difficult to verify whether IPIC offers a genuine improvement over existing, dedicated manifold alignment techniques.

**Detailed Comments:**

NA

**Justification Of The Preliminary Rating:**

While I have raised questions regarding the method's applicability to constraints regarding batch sizes in my review and compared studies, these are questions of scope rather than fundamental flaws. The core contribution—a robust, iterative framework for unpaired biological alignment—is highly significant and well-executed. This work provides a strong foundation for future research in multi-omics integration.

**Questions To Address In The Rebuttal:**

1. The evaluation does not demonstrate that the method outperforms established state-of-the-art (SOTA) algorithms specifically designed for unpaired multi-omics integration, such as 'Jointly aligning cells and genomic features of single-cell multi-omics data with co-optimal transport' (SCOT). While the authors compare against a self-constructed '1X-Matching (OT)' baseline, they do not benchmark against specialized solvers like SCOT that utilize advanced co-optimal transport strategies. Without this direct comparison, it is difficult to verify whether IPIC offers a genuine improvement over existing, dedicated manifold alignment techniques.

2. The proposed IPIC framework relies heavily on the InfoNCE loss function, which necessitates large batch sizes (e.g., N=4,096 as used in the experiments 1) to maintain a sufficient pool of negative samples for effective contrastive learning. However, this requirement presents a significant barrier for applicability to high-dimensional medical imaging (e.g., 3D MRI or CT), where GPU memory constraints typically limit training to very small batch sizes (e.g., 2–16). Consequently, the method's claim of generality across 'biological datasets' is limited, as it does not address how the contrastive mechanism can function effectively under the memory constraints inherent to 3D imaging modalities.

---

> ### Author Response · Authors · 2026-01-24
> **Thank you for your review**
>
> We thank the reviewer for their positive assessment and strong comment of our work. We appreciate your recognition of our comprehensive empirical validation across diverse datasets and your agreement that our iterative framework provides a significant and robust foundation for future multi-omics integration research.
> > Q1. do not benchmark against specialized solvers like SCOT
>
> We would like to clarify that the "1X-Matching (OT)" baseline reported in our paper is not a self-constructed heuristic, but rather the implementation of the recent state-of-the-art method "Propensity Score Alignment of Unpaired Multimodal Data" (Xi et al., NeurIPS 2024). As demonstrated in Xi et al., this propensity score-based OT approach outperforms direct manifold alignment methods like SCOT in unpaired settings. Therefore, by comparing against this method, we believe we have benchmarked against a stronger and more recent state-of-the-art baseline than SCOT.
>
> > Q2 method's claim of generality across 'biological datasets' is limited,
>
> We fully agree with the reviewer’s insight regarding the memory constraints of high-dimensional modalities like 3D MRI or CT. You are correct that contrastive learning frameworks relying on InfoNCE typically require large batch sizes (e.g., 4096) to function effectively, which presents a challenge for memory-intensive 3D data where batch sizes are often limited to single digits. We acknowledge this as a current limitation of our specific implementation compared to non-contrastive methods. In the camera-ready version, we will explicitly discuss this limitation and refine our claims regarding "biological datasets" to more accurately reflect the scope of modalities (e.g., omics, 2D pathology) for which our method is currently best suited.

---

### Official Review · Reviewer_BbUX · 2026-01-12

**Confidence:** 3
**Preliminary Rating:** 4

**Summary:**

The paper tackles a very practical issue in biological multimodal learning: when modalities can’t be measured on the same sample, you only have treatment-group labels linking datasets. The method (IPIC) builds soft pseudo-pairs within each treatment using propensity-score matching via entropic OT, and then adds an inter-treatment signal by clustering embeddings and aligning clusters across treatments. Across four datasets (two simulated-unpaired and two truly unpaired), it generally improves downstream accuracy over unpaired contrastive baselines. The takeaway is that “treatment-only supervision” can go further if you combine within-treatment matching with a cross-treatment structure prior.

**Strengths:**

The problem setting is real: a lot of wet-lab pipelines can’t give paired measurements, so designing methods that work with only group labels is valuable.
The intra-treatment OT matching is a sensible fix for the “everything in the same treatment becomes a positive” issue, this is where many naive approaches break down.
The method is modular and easy to reason about: “soft pairing” + “cluster structure” is not exotic, but it’s a clean combination.
Gains seem consistent across multiple datasets/tasks, which makes it harder to dismiss as cherry-picking.

**Weaknesses:**

The method leans on a strong assumption that the treatment-prediction distributions are comparable across modalities (so propensity scores are meaningful for matching). In real data, modality-specific artifacts can easily violate this, and the paper doesn’t really demonstrate it.
Two datasets are “unpaired by shuffling.” That’s fine for diagnostics, but it can be too nice compared to real unpaired settings where batch/protocol drift is the main headache.
Everything is done on top of precomputed embeddings. That isolates the learning objective, but it also means we don’t know how much IPIC helps end-to-end, or how sensitive it is to embedding quality.
The clustering term feels heuristic: KMeans with a chosen  K can be unstable, and I’d like to see more on when it fails or becomes noisy.

**Detailed Comments:**

I’d really like one “robustness” experiment: add modality-specific confounding/batch shifts and see whether the OT matching still helps or starts matching the wrong things.
Please report clustering stability (e.g., variability across seeds) and whether the inter-treatment pseudo-labels drift a lot during training.
Runtime/scaling: OT per treatment could become expensive; a short scaling plot or complexity discussion would help.

**Justification Of The Preliminary Rating:**

accept because this is a common real-world constraint, and the method is a reasonable, nontrivial way to squeeze useful supervision out of treatment labels. The empirical improvements look consistent, and the “soft matching within treatment” piece in particular seems like a meaningful contribution. What holds me back from a stronger score is mostly external validity: the approach depends on assumptions that can break under batch/protocol drift, and the evaluation is partly in a “nice” simulated-unpaired regime. If the authors can clarify robustness/failure modes, I’d feel better about the general claim.

**Questions To Address In The Rebuttal:**

How sensitive is performance to violations of the propensity-score assumption (e.g., treatment is much easier to predict in one modality)?
Can you test a more realistic unpaired setting (cross-batch / cross-lab split) rather than only shuffling within treatment?
Does IPIC still help if upstream embeddings are weaker, or if one modality’s encoder is much noisier?

---

> ### Author Response · Authors · 2026-01-24
> **Thank you for your review**
>
> We thank the reviewer for their thoughtful and detailed review. We appreciate your recognition of the practical importance of learning from unpaired biological data where group labels are the only link, and we are glad you found our Intra-treatment OT matching to be a sensible solution to the positive-pair problem. We also value your observation that our method’s modularity and consistent empirical gains make it a robust approach.
>
> > How sensitive is performance to violations of the propensity-score assumption (e.g., treatment is much easier to predict in one modality)?
>
> We acknowledge the concern regarding the assumption that treatment prediction distributions are comparable. Theoretically, if this assumption held perfectly, the "One-time Matching" baseline would likely yield the best performance, as it strictly relies on these scores. However, our results show that IPIC (which matches iteratively) consistently outperforms One-time Matching. This suggests that IPIC is robust to violations of this assumption. For example, in the ST dataset, the treatment is significantly easier to predict via the image (pathology) modality than the transcriptomics modality, yet IPIC successfully recovers alignment and improves downstream performance despite this asymmetry.
>
> > Can you test a more realistic unpaired setting (cross-batch / cross-lab split) rather than only shuffling within treatment?
>
> We would like to clarify that COMP and GKO are authentic, naturally unpaired, and cross-batch datasets derived from separate experiments, rather than simulated environments. Regarding BIRFL, while it is shuffled, it is designed to be challenging rather than "nice": we introduced 10% "no treatment effect," 10% "combined effect," and 10% "other treatment effect" samples specifically to simulate the noisy, non-ideal conditions found in real biological data.
>
> > Does IPIC still help if upstream embeddings are weaker, or if one modality’s encoder is much noisier?
>
> As shown in Table 7, we include a "Raw Features" baseline. Comparing this against IPIC demonstrates that our method significantly improves predictive power over the raw embeddings, regardless of the initial quality. While a weaker upstream model would naturally lower the absolute baseline score, our experiments with different embedding models on the fluorescent single-cell images (GKO and COMP) consistently showed that IPIC improves upon the base encoder. We view this work as focusing on how to learn between modalities rather than optimizing the encoder architecture itself. However, we agree that a dedicated robustness test varying different encoders is a valuable suggestion and will leave this extensive benchmarking for future work.

---

### Meta-Review · Area_Chair_7t1z · 2026-02-02

**Recommendation:** Accept (Oral)
**Confidence:** 5

**Metareview:**

All reviewers are leaning towards accepting this paper with two recommendations rating this work rather highly. The authors additionally addressed remaining concerns during the discussion/rebuttal phase. I believe this paper is a clear accept.

---

### Decision · Program_Chairs · 2026-02-13

Accept (Poster)